# Identification of a *c*-type heme oxygenase and its function during acclimation of cyanobacteria to nitrogen fluctuations

Zhaoxing Ran [1,2], Zhenyu Du[2], Gengkai Miao[2], Mei Zheng[2], Ligang Luo[2], Xiaoqin Pang[2], Lanzhen Wei [2✉], Dezhi Li [1,3,4,5✉] & Weimin Ma [2✉]

The mechanisms of acclimating to a nitrogen-fluctuating environment are necessary for the survival of aquatic cyanobacteria in their natural habitats, but our understanding is still far from complete. Here, the synthesis of phycobiliprotein is confirmed to be much earlier than that of photosystem components during recovery from nitrogen chlorosis and an unknown protein Ssr1698 is discovered to be involved in this synthetic process. The unknown protein is further identified as a *c*-type heme oxygenase (*c*HO) in tetrapyrrole biosynthetic pathway and catalyzes the opening of heme ring to form biliverdin IXα, which is required for phycobilin production and ensuing phycobiliprotein synthesis. In addition, the *c*HO-dependent phycobiliprotein is found to be vital for the growth of cyanobacterial cells during chlorosis and regreening through its nitrogen-storage and light-harvesting functions, respectively. Collectively, the *c*HO expressed preferentially during recovery from nitrogen chlorosis is identified in photosynthetic organisms and the dual function of this enzyme-dependent phycobiliprotein is proposed to be an important mechanism for acclimation of aquatic cyanobacteria to a nitrogen-fluctuating environment.

[1] School of Ecological and Environmental Sciences, East China Normal University, 200241 Shanghai, China. [2] College of Life Sciences, Shanghai Normal University, 200234 Shanghai, China. [3] Key Laboratory of Urbanization and Ecological Restoration of Shanghai, 200241 Shanghai, China. [4] Institute of Eco-Chongming (IEC), 20 Cuiniao Rd, Chenjia Zhen, Chongming, 202162 Shanghai, China. [5] Technology Innovation Center for Land Spatial Eco-restoration in Metropolitan Area, Ministry of Natural Resources, 3663 N. Zhongshan Road, 200062 Shanghai, China. ✉email: weilz@shnu.edu.cn; dzli@des.ecnu.edu.cn; wma@shnu.edu.cn

Cyanobacteria (also referred to as blue-green algae) are a diverse group of predominantly aquatic photosynthetic organisms and account for up to 50% of the $N_2$ fixed as dissolved nitrogen[1,2] and more than 35% of global $CO_2$ fixation that takes place on Earth[3–6]. It has been widely accepted that the aquatic environment that they inhabit is overwhelmed with the nutrient stress of nitrogen fluctuations[7–9]. Very fortunately, numerous mechanisms of acclimating to the nitrogen-fluctuating environment have been developed in aquatic cyanobacterial cells and they can jointly alleviate such stress and guarantee an efficient nitrogen storage and photosynthetic $CO_2$ fixation[10–12].

Chlorosis of non-nitrogen-fixing cyanobacteria is induced by nitrogen starvation and their regreening is triggered after the nutrient becomes available[13]. In this process, glycogen is accumulated efficiently in nitrogen-starved cells and its breakdown is of great importance for rapid regreening of chlorotic cells as their respiratory substrate with the aid of glycogen phosphorylase[12–14]. In contrast, the peripheral light-harvesting antenna phycobilisome of cyanobacteria and their photosystems are synthesized in nitrogen-replete cells and their degradation triggered by nitrogen starvation is vital to provide additional amino-acids and decrease excitation pressure[15–17]. In addition, cyanophycin, a co-polymer of aspartate and arginine, is also synthesized under nitrogen-replete conditions as a nitrogen reservoir and is utilized under nitrogen limitation as nitrogen source with the cooperation of cyanophycinase, isoaspartyl dipeptidase, and arginine dihydrolase (ArgZ)[18]. Recently, an active ornithine–ammonia cycle (OAC) is identified in cyanobacteria and the OAC allows for efficient storage of cyanophycin in nitrogen-replete cells and its rapid utilization in nitrogen-starved cells[11]. Collectively, much progress has been made in understanding the mechanisms underlying the acclimation of aquatic cyanobacteria to nitrogen fluctuations, but our understanding is still far from complete.

The recently reported proteomic data indicated that synthesis of phycobiliprotein (PBP) is much earlier than that of photosystem components after the nitrate is added to chlorotic cells[19] and heme oxygenase (HO) is required for the early synthesis of PBP through catalyzing the opening of the heme ring with the release of iron to form biliverdin[20]. In cyanobacteria, two canonical types of $b$-type heme oxygenase, $b$HO-1 and $b$HO-2, have been identified[21,22], but the $c$-type heme oxygenase ($c$HO) remains a mystery, although it has been identified in bacteria[23,24]. This reminds us that there may be an unknown mechanism underlying the acclimation of aquatic cyanobacteria to nitrogen fluctuations.

Inspired by this possibility, we first confirmed that the synthesis of PBP during recovery from nitrogen chlorosis was much earlier than that of photosystem components and discovered that an unknown protein Ssr1698 was involved in this synthetic process. Then, the unknown protein was identified as a $c$-type heme oxygenase ($c$HO) in the tetrapyrrole biosynthetic pathway and catalyzed the opening of heme ring to form biliverdin IXα, which was required for phycobilin production and ensuing PBP synthesis. Finally, the $c$HO-dependent PBP was found to be vital for the growth of cells during chlorosis and regreening through its nitrogen-storage and light-harvesting functions, respectively, and its dual function was proposed to be an important mechanism for acclimation of aquatic cyanobacteria to a nitrogen-fluctuating environment.

## Results

### The early synthesis of phycobiliprotein during recovery from nitrogen chlorosis.

Consistent with the recently reported proteomic data[19], our protein blot results confirmed that the synthesis of phycobiliprotein (PBP) was much earlier than that of

components of PSI and PSII after the nitrate was added to chlorotic cells (Fig. 1a, b, f and Supplementary Figs. 1 and 2). This finding was consolidated by the data of assembly and function of PBP, PSI, and PSII components after the nitrate was added to chlorotic cells (Fig. 1c–f and Supplementary Fig. 2). Collectively, apart from the light-harvesting function, it is logical to hypothesize that the early synthesis of PBP during recovery from the nitrogen chlorosis may also serve as a nitrogen reservoir and function in ensuing nitrogen-starved cells.

### Involvement of an unknown protein Ssr1698 in the early synthesis of phycobiliprotein during recovery from nitrogen chlorosis.

To test this hypothesis, a comparative transcriptomic analysis was conducted after the nitrate was added to chlorotic cells for 4 h (i.e., at an early stage) or not. Our transcriptomic data showed that the 168 genes were upregulated after the nitrate was added to chlorotic cells and they include many apo-PBP-related genes that encode rod subunits, core subunits, rod linker, rod-core linker, and core-membrane linker, but rarely include those known genes that encode catalytic enzymes in tetrapyrrole biosynthetic pathway for phycobilin production (Supplementary Figs. 3 and 4), which is required for PBP synthesis. It appears plausible that certain unknown catalytic enzymes are involved in the tetrapyrrole biosynthetic pathway for the synthesis of PBP during recovery from nitrogen chlorosis.

To test this possibility and explore the possible novel mechanisms underlying the acclimation of cyanobacteria to the nitrogen-fluctuating environment, the top five upregulated unknown genes in our transcriptomic data were selected as candidates for PBP synthesis to construct their mutants and measure their levels of accumulated PBP after the nitrogen becomes available (Supplementary Fig. 5). As deduced from the accumulated PBP levels, we fortunately found that deletion of $ssr1698$ gene suppressed the synthesis of PBP and ensuing its assembly and function after the nitrate was added to chlorotic cells for 12 h (Fig. 2a–f and Supplementary Fig. 5). Taken together, we propose that the unknown protein Ssr1698 is involved in the early synthesis of PBP during recovery from nitrogen chlorosis.

### Identification of Ssr1698 as a $c$-type heme oxygenase.

Ssr1698 is a 96-amino acid protein with a DUF2470 domain and is conserved only in cyanobacteria (Supplementary Fig. 6), but its DUF2470 domain is identified also in bacterial HugZ and ChuZ (Fig. 3a and Supplementary Fig. 7) and is predicted to provide the interacting interface with heme[25,26]. To test this prediction, we carried out the heme titration of the purified tag-less Ssr1698 protein (Supplementary Fig. 8) in the presence of BSA, which is used to exclude the possibility of nonspecific binding of heme[27,28]. The results demonstrated the specific binding of $c$-type heme, but not $b$-type heme, to Ssr1698 (Fig. 3b and Supplementary Fig. 9). Furthermore, we found that with an increase in the concentration of $c$-type heme, the Soret peak exhibits a progressive growth, reaching its maximum at a 1:2 binding stoichiometry of $c$-type heme to Ssr1698 (Fig. 3c). The binding stoichiometry of $c$-type heme to Ssr1698 suggested that two Ssr1698 proteins as a dimer share a heme molecule (Fig. 3c), as shown in the predicted structure (Fig. 3e). This dimeric state of Ssr1698 was supported by the data of size-exclusion chromatogram (Supplementary Fig. 8) and CN-PAGE (Fig. 3d). Considering the heme-binding property of Ssr1698 protein and its association to PBP synthesis, we hypothesize that this protein, as a potential heme oxygenase, is involved in tetrapyrrole biosynthetic pathway for biliverdin production and subsequent PBP synthesis.

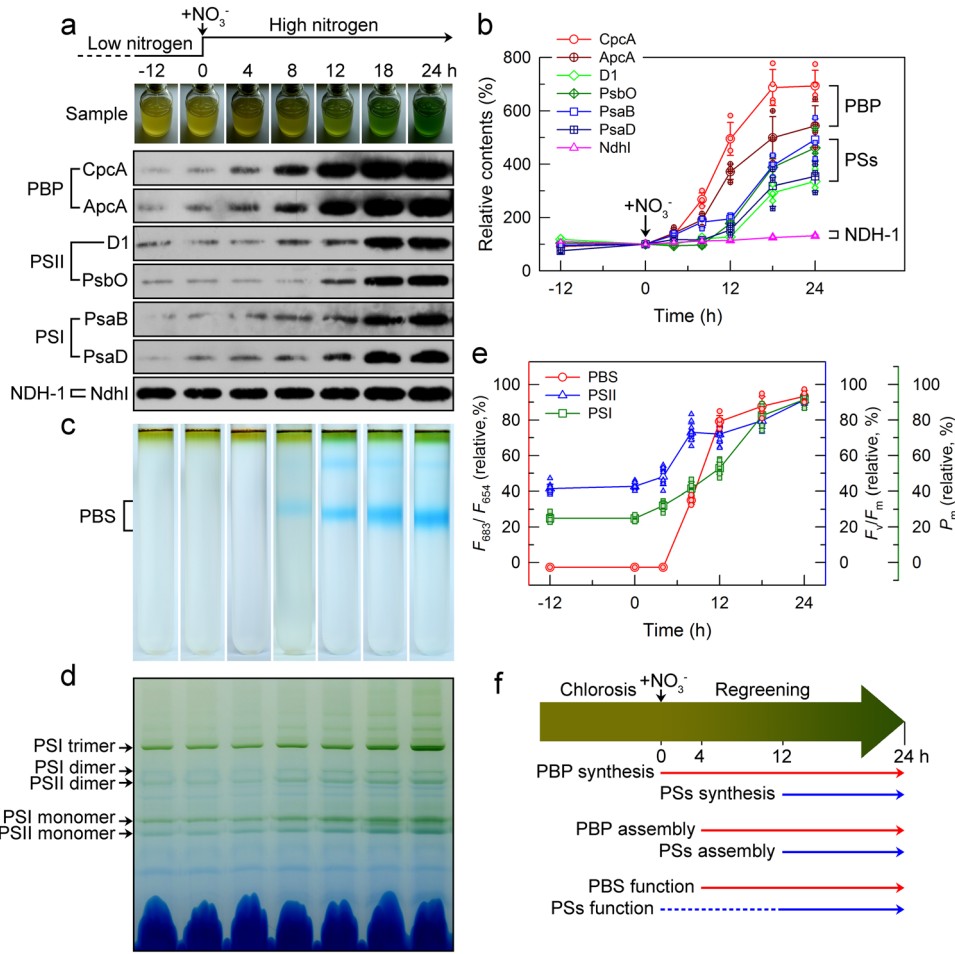

**Fig. 1 Synthesis, assembly, and function of phycobiliprotein and both photosystems during recovery from nitrogen chlorosis. a** Time-course synthesis profile of phycobiliprotein (PBP) and both photosystems (PSs) after the nitrate is added to chlorotic cells. Total protein corresponding to $3 \times 10^7$ cells was loaded onto each lane and NdhI was used as a sample loading control. **b** Quantification of PBP (CpcA and ApcA, $n = 4$), PSs (D1, PsbO, PsaB, and PsaD, $n = 3$) and NDH-1 (NdhI, $n = 3$) synthesis levels using ImageJ. **c, d** Time-course assembly profiles of phycobilisome (PBS) isolated by sucrose density gradient centrifugation (**c**) and PSs isolated by BN-PAGE (**d**) after the nitrated is added to chlorotic cells. **e** Time-course function profiles of PBS and PSs after the nitrate is added to chlorotic cells. Red lines (circle): PBS ($n = 6$); blue lines (triangle up): PSII ($n = 12$); green lines (square): PSI ($n = 12$). **f** A model schematically represents the temporal sequence of synthesis, assembly and function of PBP and PSs during recovery from nitrogen chlorosis.

It has been widely accepted that heme oxygenase catalyzes the opening of the heme ring with the release of iron to form biliverdin, which is required for phycobilin production and subsequent PBP synthesis[20]. Two canonical types of *b*-type heme oxygenase, *b*HO-1 and *b*HO-2, have been identified in a unicellular cyanobacterium *Synechocystis* sp. strain PCC 6803 (hereafter referred to as *Synechocystis* 6803) and preferentially cleave-free *b*-type heme[21,22]. Another common form of heme found in nature is *c*-type heme, which is covalently linked to proteinaceous cysteine residues[23]. In photosynthetic organisms, however, the *c*-type heme oxygenase (*c*HO) remains a mystery.

To test whether the unknown protein Ssr1698 works as a new *b*HO-3 or mysterious *c*HO, we determined the enzymatic reaction in the presence of *b*-type heme and *c*-type heme, respectively. The experimental data clearly showed that Ssr1698 lacked any observable reactivity with *b*-type heme (Supplementary Fig. 10), but had the ability to degrade the *c*-type heme (Fig. 4a). Moreover, with the aid of high-performance liquid chromatography (HPLC), mass spectrometry and ferrozine assay analyses, the biliverdin IXα molecule, 11-amino acid peptide and free iron were identified in the reaction product catalyzed by the Ssr1698 protein with *c*-type heme as a substrate (Fig. 4b–f and Supplementary Figs. 11 and 12). Collectively, we propose that

the Ssr1698 protein works as a mysterious *c*HO and is required for biliverdin production and subsequent PBP synthesis. To the best of our knowledge, this is the first study to identify the *c*HO in photosynthetic organisms.

In cyanobacteria, the photosynthetic *c*HO enzyme can catalyze the cleavage of thioether bond, since we identify an 11-amino acid peptide in the reaction product catalyzed by the enzyme with *c*-type heme as a substrate (Fig. 4d and Supplementary Fig. 12). The enzyme can also catalyze the opening of the heme ring at the α-meso position to form biliverdin IXα (Fig. 4c). The opening is supported by the results of studies showing that the mass spectrometry fragmentation pattern of the 583.3 *m/z* peak of the reaction product catalyzed by the *c*HO enzyme with *c*-type heme as a substrate is consistent with that of biliverdin IXα, a standard reference sample (Supplementary Fig. 13). The opening is also supported by the results that the degradation of *c*-type heme is significantly suppressed after the addition of biliverdin IXα to the reaction system catalyzed by the *c*HO enzyme (Supplementary Fig. 14)[29]. Undoubtedly, this catalytic mechanism of cyanobacterial *c*HO is different from that of bacterial *c*HO. This difference may be caused by their different domains that cyanobacterial *c*HO contains DUF2470 domain (Fig. 3a) and bacterial *c*HO contains β-barrel domain[23].

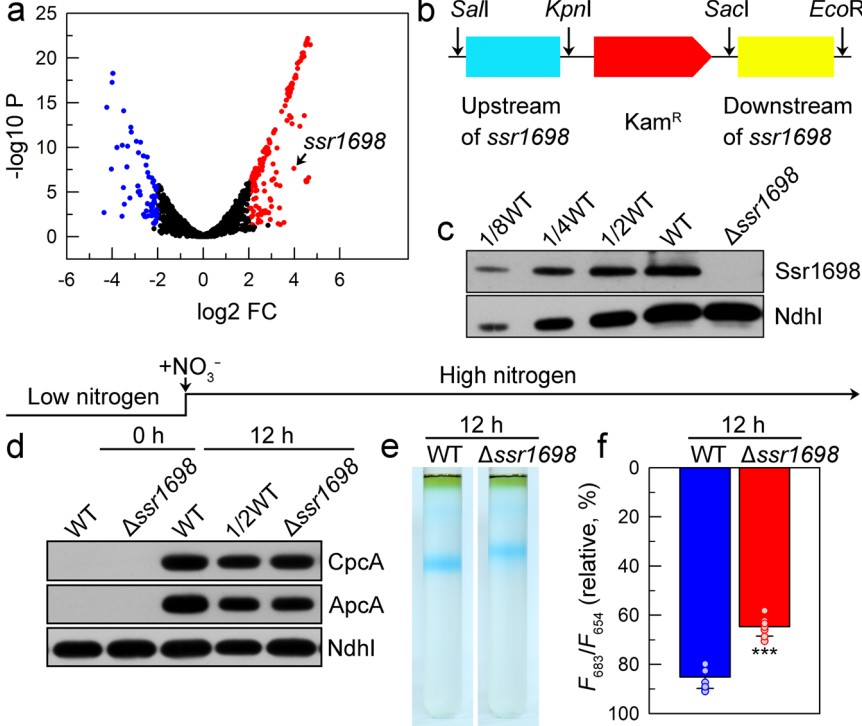

**Fig. 2 Ssr1698 is of great importance for synthesis, assembly, and function of phycobiliprotein during recovery from nitrogen chlorosis. a** A comparative transcriptomic analysis was conducted after the nitrate was added to chlorotic cells for 4 h or not, and a top one upregulated unknown gene *ssr1698* (see arrow) in our transcriptomic data was selected as a candidate for synthesis, assembly, and function of phycobiliprotein after the nitrogen becomes available. **b** Construction of the plasmid used to generate the *ssr1698*-deletion mutant (Δ*ssr1698*). **c** Western analysis of Ssr1698 from the total protein of the WT (including indicated serial dilutions) and Δ*ssr1698* strains. Total protein corresponding to 1 µg chlorophyll *a* was loaded onto each lane, and NdhI was detected as a loading control. **d–f** Deletion of *ssr1698* impairs the synthesis (**d**), assembly (**e**) and function (**f**) of PBP after the nitrate is added to chlorotic cells for 12 h. Error bars denote the standard deviations of eight independent measurements ($n = 8$); ***Represents $P < 0.0001$ ($P = 1.5 \times 10^{-7}$).

**Dual function of the *c*HO-dependent phycobiliprotein during acclimation of cyanobacteria to nitrogen fluctuations.** To unravel the function of *c*HO-dependent PBP produced during recovery from nitrogen chlorosis, the growth of *Synechocystis* 6803 cells was measured after the nitrate was added to chlorotic cells or the nitrate was removed from the regreening cells. Consistent with the previously reported results of truncated peripheral light-harvesting antenna[30], decreased synthesis of PBP caused by the inactivation of Ssr1698 (Fig. 2d–f) evidently retarded the growth of *Synechocystis* 6803 cells under conditions of low light (Supplementary Fig. 15a) and even growth light (Fig. 5a), but rarely under high light (Supplementary Fig. 15b) after the nitrate was added to chlorotic cells. This confirms our expectation that the *c*HO-dependent PBP is involved in light-harvesting function during recovery from nitrogen chlorosis.

Compared to the components of PSI and PSII, the synthesis of PBP has been demonstrated to be much earlier during recovery from nitrogen chlorosis (Fig. 1a, b)[19]. Inspired by the fact that cyanophycin is synthesized under nitrogen repletion as nitrogen reservoir and is utilized under nitrogen limitation as a nitrogen source[11,18], it is logical to hypothesize that the early synthesized PBP may be utilized under nitrogen deficiency as a nitrogen source. To test this hypothesis, the growth of *Synechocystis* 6803 cells was measured after the nitrate was removed from the regreening cells. The results indicated that decreased synthesis of PBP caused by the inactivation of Ssr1698 in regreening cells (Fig. 2d–f) considerably retarded the growth of cells under subsequent nitrogen starvation (Fig. 5b). Therefore, we propose that, besides the light-harvesting function, the *c*HO-dependent PBP is also involved in constructing nitrogen reservoir during

recovery from nitrogen chlorosis and function in ensuing nitrogen-starved cells as a nitrogen source.

To reveal the importance of the dual function of *c*HO-dependent PBP during the acclimation of cyanobacteria to nitrogen fluctuations, we monitored the growth of wild-type (WT) *Synechocystis* 6803 and its *ssr1698*-deletion mutant cells during repeated cycles of nitrogen deprivation followed by replenishment treatments. Compared with the WT, the inhibition degree of cell growth caused by the inactivation of Ssr1698 gradually increased with the increase in repeated cycles of nitrogen oscillation (Fig. 5c, d), but not in repeated cycles of nitrogen repletion (Supplementary Fig. 16), being consistent with the results that Ssr1698 was expressed under nitrogen oscillation (preferentially during recovery from nitrogen chlorosis; Fig. 2a) and not under nitrogen repletion. It is worthy of note that, in the third cycle of nitrogen oscillation, the inactivation of Ssr1698 has led to more than 40% decrease in the cell dry weight (Fig. 5c, d). Collectively, we propose that the dual function of the *c*HO-dependent PBP is of great importance during the acclimation of cyanobacteria to a nitrogen-fluctuating environment.

**The *c*HO homologs are geographically widespread.** To see whether the dual-function mechanism of the *c*HO-dependent PBP operates also in oceanic environment during acclimation of aquatic cyanobacteria to a nitrogen-oscillation condition, the complete protein sequence of Ssr1698 was queried against the Tara Oceans database[31]. By clicking the size fractionation range of cyanobacterial cells and using stringent *E* values, 164 hits were obtained for cyanobacteria (*E* value < 1.0 E$^{-10}$). Ssr1698 homolog sequences were found predominantly in the surface water layer

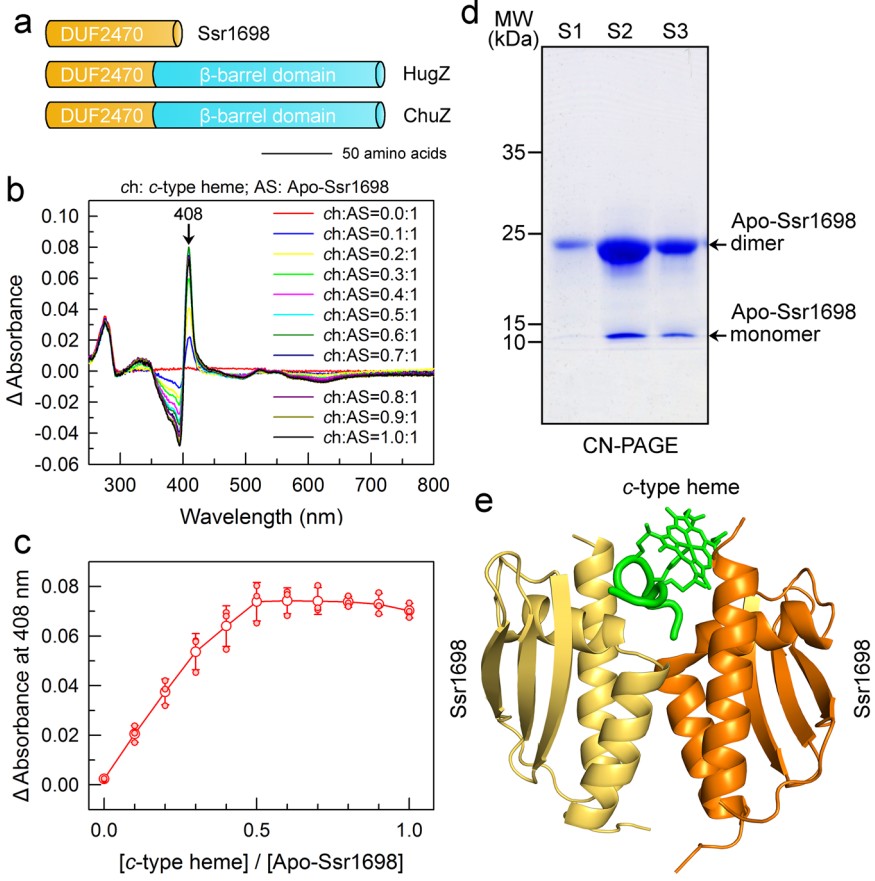

**Fig. 3 The binding of apo-Ssr1698 with *c*-type heme. a** Ssr1698 shares the DUF2470 domain with bacterial HugZ and ChuZ, which are predicted to provide the interacting interface with heme. **b** The MP-11 (*c*-type heme) titration of apo-Ssr1698 (10 μM) as monitored by the difference absorption spectra in PBS buffer (pH 7.4) and 20 μM BSA. The positive characteristic absorption peaks indicate the binding of apo-Ssr1698 with *c*-type heme. **c** The titration curve observed at 408 nm for apo-Ssr1698 with *c*-type heme. Error bars indicate the standard deviations of three independent replicates ($n = 3$). **d** The oligomerization state of purified tag-less apo-Ssr1698 is determined by CN-PAGE. Enzyme digestion and purification of S1, S2, and S3 are shown in Supplementary Fig. 8. **e** A structural model represents the binding of apo-Ssr1698 with *c*-type heme. Apo-Ssr1698 structure was predicted by AlphaFold, and docking of its dimer with *c*-type heme was predicted by AutoDockVina.

(SRF) and deep chlorophyll maximum layer (DCM) but, less abundantly, in the mesopelagic zone (MES), and were almost equally distributed among the latitudes (Supplementary Fig. 17). By comparison, recurrent nitrate oscillation and abundant cyanobacteria occur in the SRF and DCM but, less in the MES[32,33]. Collectively, we propose that the dual-function mechanism of *c*HO-dependent PBP is vital for acclimation of marine cyanobacteria to nitrogen-oscillation layers.

## Discussion

The opening of the heme ring to form biliverdin IXα catalyzed by heme oxygenase is a necessary step to produce phycobilin in tetrapyrrole biosynthetic pathway and subsequently to synthesize PBP (Supplementary Fig. S4)[20]. Although bacterial *b*HOs ChuZ and HugZ and cyanobacterial *c*HO share a common DUF2470 domain (Supplementary Fig. 7), they only have a weak sequence similarity (Supplementary Fig. 18), increasing the difficulty of the discovery of *c*HO in photosynthetic organisms. In addition, the results of previous studies and this study indicated that the *b*HO-1, *b*HO-2, and *c*HO genes *sll1184*, *sll1875* and *ssr1698* were preferentially expressed under conditions of standard growth, hypoxia and regreening, respectively (Supplementary Fig. 19)[34–36]. It appears plausible that the *c*-type heme, a substrate of *c*HO enzyme, released from the degradation of various *c*-type heme-binding proteins under nitrogen deficiency induces the expression of *c*HO enzyme

during subsequent recovery from nitrogen chlorosis (Supplementary Fig. 19)[19,37–39]. This may be a reason why the photosynthetic *c*HO enzyme can be first identified in this study.

It has been widely accepted that the PBP can be assembled into phycobilisome supercomplex as peripheral light-harvesting antenna[40] and account for up to 60% of the total soluble cell protein as nitrogen reservoir[41]. The results of this study indicated that the amount of PBP was reduced to about half the wild-type level in the *c*HO-deletion mutant after the nitrate was added to chlorotic cells for 12 h (Fig. 2 and Supplementary Fig. 5). Therefore, we hypothesize that the *c*HO-dependent PBP is involved in light-harvesting and nitrogen reservoir functions during acclimation of aquatic cyanobacteria to nitrogen fluctuations.

Our data reported here confirm our hypothesis and unravel a dual-function mechanism of *c*HO-dependent PBP during acclimation of aquatic cyanobacteria to a nitrogen-fluctuating environment that occurs frequently in their natural habitats. The *c*HO-dependent PBP produced during recovery from nitrogen chlorosis is involved in light-harvesting function during the recovery process (left, Fig. 6) and nitrogen reservoir function during nitrogen deficiency (right, Fig. 6) and as a result, in the third cycle of nitrogen oscillation, inactivation of Ssr1698 has led to more than 40% decrease in the cell dry weight (Fig. 5). This indicates that the *c*HO-dependent PBP produced during

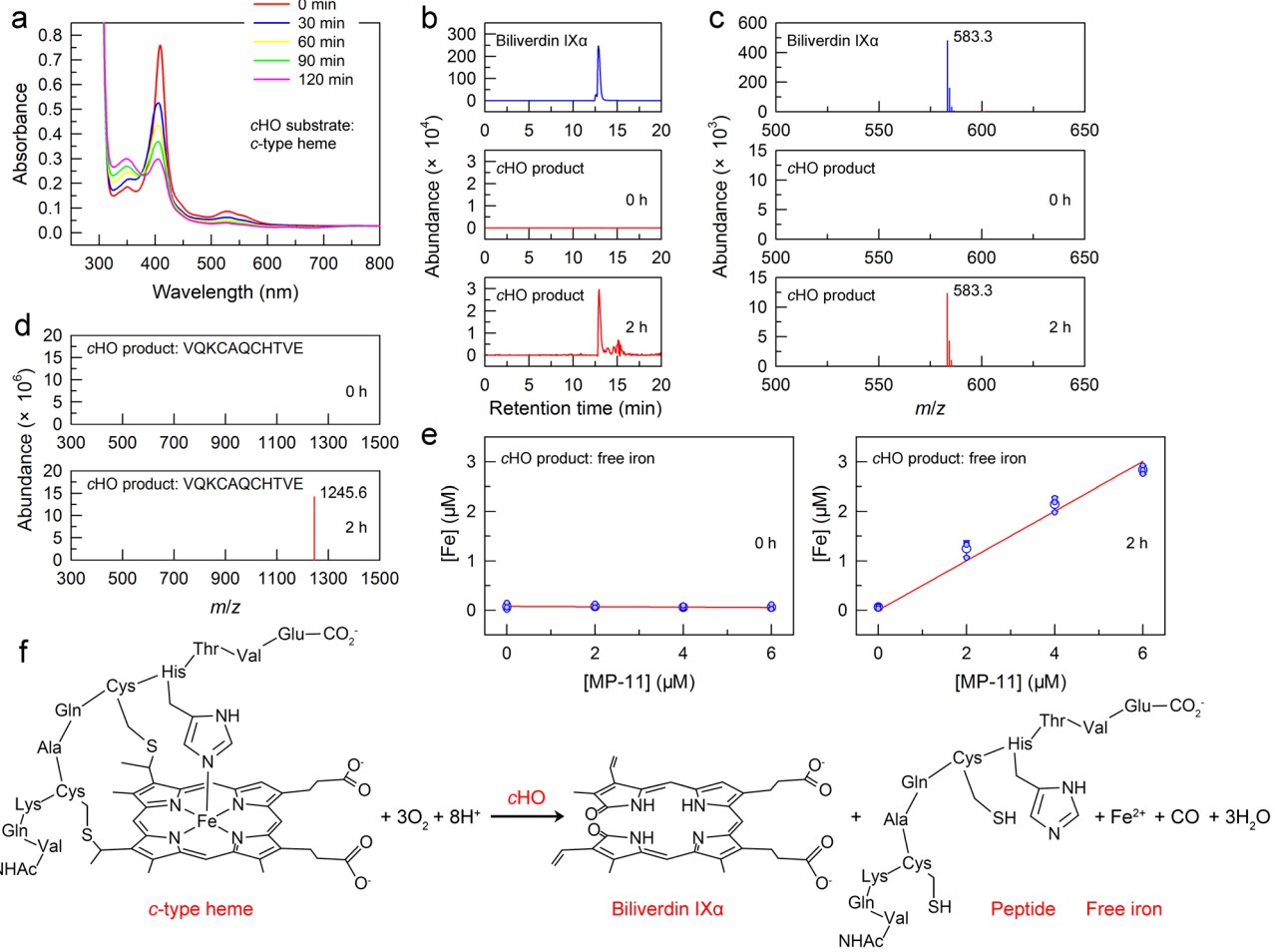

**Fig. 4 Ssr1698 catalyzes the reaction of *c*-type heme to biliverdin IXα. a** The degradation assay of MP-11 (*c*-type heme) with Ssr1698. The decline in the characteristic Soret peak of heme at 408 nm and its shoulder peaks between 500 nm and 600 nm indicates that Ssr1698 degraded the *c*-type heme, a substrate for Ssr1698. **b–e** HPLC (**b**), MS spectra (**c**, **d**) and ferrozine assay (**e**) data indicate the presence of a biliverdin IXα molecule, 11-amino acid peptide and free iron in the reaction product catalyzed by Ssr1698 (*c*HO) for 2 h and not for 0 h. Error bars indicate the standard deviations of three independent replicates ($n = 3$). **f** An equation for the reaction of *c*-type heme to biliverdin IXα catalyzed by *c*HO.

regreening is of great importance for the growth of aquatic cyanobacteria to a nitrogen-fluctuating environment.

Over the past decade, a significant achievement has been made in identifying the working mechanisms during acclimation of aquatic cyanobacteria to the nitrogen-fluctuating environment[11–18]. Unraveling these mechanisms will certainly help deepen our understanding of the reason why cyanobacteria account for a large percentage of nitrogen storage[41] and photosynthetic $CO_2$ fixation on Earth[3–6]. The results of this study indicated that the *c*HO-dependent PBP produced during recovery from nitrogen chlorosis is another important working mechanism of acclimating of aquatic cyanobacteria to nitrogen fluctuations.

Recently, Zhang et al.[11] reported that cyanophycin also functions as a nitrogen reservoir and its defects caused by inactivation of arginine dihydrolase gene (*argZ*) retarded the growth of cyanobacterial cells during their acclimation to nitrogen fluctuations. To test whether the previously reported cyanophycin and the present *c*HO-dependent PBP produced during recovery from nitrogen chlorosis jointly serve as a nitrogen reservoir and provide nitrogen source in ensuing nitrogen-starved cells, we constructed a Δ*ssr1698*/*argZ* double-mutant strain (Supplementary Fig. 20). Our data indicated that the growth of the double mutant Δ*ssr1698*/*argZ* was much slower than that of its respective single mutants, Δ*ssr1698* and Δ*argZ*, in repeated cycles of nitrogen oscillation (Fig. 5 and Supplementary Fig. 20). This indicates that

the dual function of *c*HO-dependent PBP is actually not redundant and is an independent working mechanism of acclimating of aquatic cyanobacteria to the nitrogen-fluctuating environment.

Collectively, to the best of our knowledge, this is the first study to identify the *c*HO enzyme in photosynthetic organisms, which is preferentially expressed during recovery from nitrogen chlorosis and is a key step to produce phycobilin in tetrapyrrole biosynthetic pathway and subsequently to synthesize PBP. This enzyme-dependent PBP is further found to be vital for the growth of cyanobacterial cells during chlorosis and regreening through its nitrogen-storage and light-harvesting functions, respectively, and its dual function is proposed to be an important mechanism for acclimation of aquatic cyanobacteria to a nitrogen-fluctuating environment.

## Methods

**Culture conditions.** *Synechocystis* sp. strain PCC 6803 (hereafter referred to as *Synechocystis* 6803) glucose tolerant strain (the wild-type) and its mutants, Δ*ssr1698*, Δ*sll0253*, Δ*ssl0483*, Δ*sll1654*, Δ*ssr1528*, Δ*sll1336* (*argZ*) and Δ*ssr1698*/*argZ* were cultured at 30 °C in BG-11 medium[42] buffered with Tris-HCl (5 mM, pH 8.0) and bubbled with 2% (*v*/*v*) $CO_2$ in air. For the growth of cells under nitrogen-deficient conditions, the concentration of nitrate in BG-11 medium was reduced from 17.6 mM to 0.35 mM. For

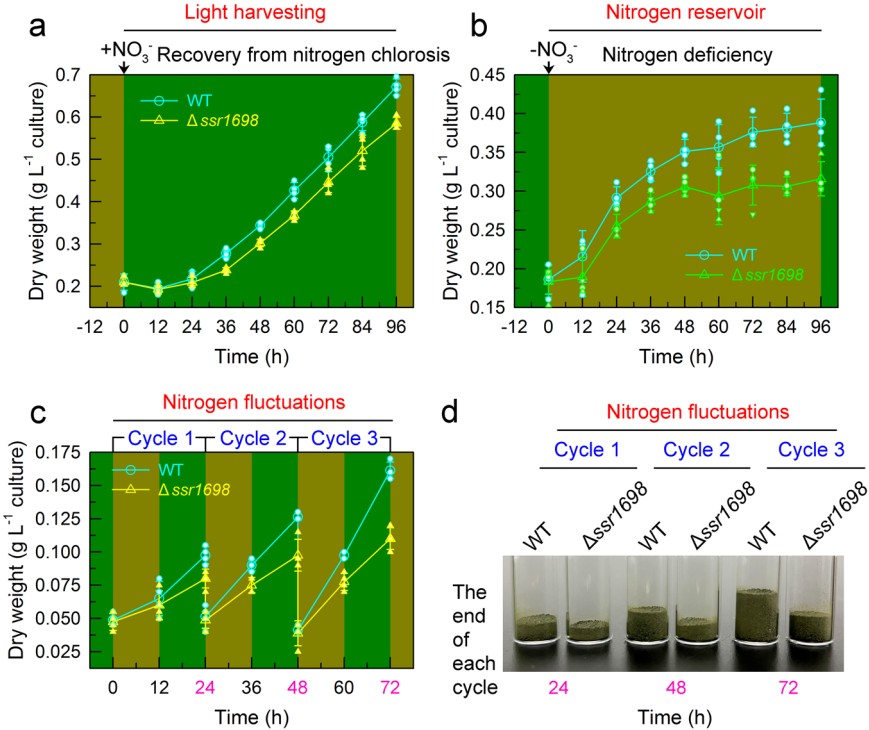

**Fig. 5 Dual function of *c*HO-dependent phycobiliprotein during acclimation of growth light-illuminated cyanobacteria to a nitrogen-fluctuating environment.** **a**, **b** The *c*HO-dependent phycobiliprotein is important for the cell growth via assembling into phycobilisome (light-harvesting function) after the nitrate is added to nitrogen-starved cells (**a**) and decomposing into nitrogen source (nitrogen-storage function) after the nitrate is removed from the culture medium (**b**). Error bars denote the standard deviations of four independent measurements (*n* = 4). **c**, **d** The *c*HO-dependent phycobiliprotein is of great importance for the growth of cells in repeated cycles of nitrogen deprivation followed by replenishment treatments. Error bars denote the standard deviations of four independent measurements (*n* = 4).

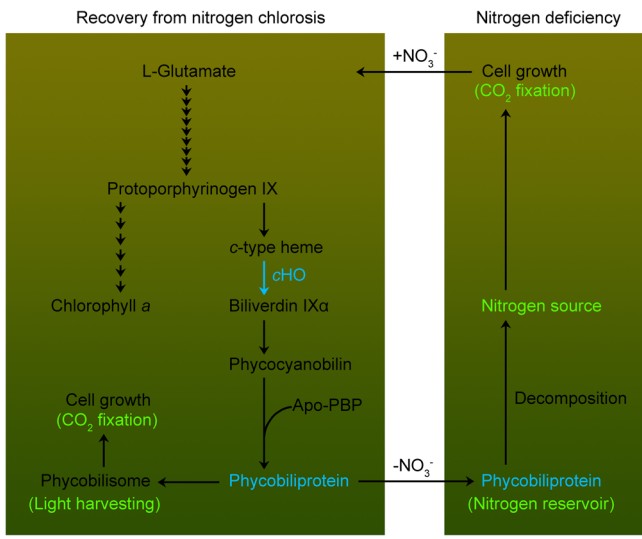

*c*HO, *c*-type heme oxygenase; Apo-PBP, Apo-phycobiliprotein

**Fig. 6 A model schematically represents the dual function of *c*HO-dependent phycobiliprotein during the acclimation of aquatic cyanobacteria to nitrogen fluctuations.** In the tetrapyrrole biosynthetic pathway, *c*HO catalyzes the reaction of *c*-type heme to biliverdin IXα (left side, see blue arrow), which is of great importance for the synthesis, assembly and function of phycobiliprotein during recovery from nitrogen chlorosis. The *c*HO-dependent phycobiliprotein is important for the cell growth via assembling into phycobilisome as light-harvesting function after the nitrate is added to chlorotic cells (left side) and decomposing into nitrogen source as nitrogen-storage function after the nitrate is removed from the culture medium (right side).

the recovery of cells from nitrogen chlorosis, the nitrate was added to nitrogen-starved cultures to a final concentration of 17.6 mM. Continuous illumination was provided by fluorescence lamps at 5, 40, or 200 μmol photons m$^{-2}$ s$^{-1}$. The mutant strains were routinely grown in the presence of appropriate antibiotics.

**Transcriptomic sequencing and bioinformatics analysis.** The nitrate was added to chlorotic cells of wild-type *Synechocystis* 6803 for 4 h or not and total RNA was extracted using the mirVana miRNA Isolation Kit (Ambion, Austin, TX, USA) according to the manufacturer's protocol. RNA integrity was assessed using the Agilent 2100 Bioanalyzer (Agilent Technologies, Santa Clara, CA, USA), and samples with an RNA Integrity Number (RIN) greater than 7 were used for subsequent analysis. The libraries were constructed using TruSeq Stranded Total RNA with Ribo-Zero Gold according to the manufacturer's instructions, and transcriptome sequencing was performed using the Illumina sequencing platform by Shanghai OE Biotech. Co. Ltd (Shanghai, China). Differentially expressed genes (DEGs) were identified with DESeq, using *P* value < 0.05 and fold change > 4 or < −4 as the thresholds. Cluster analysis of DEGs was performed according to the gene function annotation from CyanoBase (http://genome.microbedb.jp/cyanobase/).

**Real-time quantitative PCR analysis.** Total RNA used for real-time quantitative PCR (RT-qPCR) analysis was derived from the chlorotic cells of wild-type *Synechocystis* 6803 that were added by nitrate for 4 h or not. Extracted total RNA was reverse transcribed to cDNA using PrimeScript$^{TM}$ RT reagent Kit (Perfect Real Time) (TAKARA, Otsu, Japan) according to the manufacturer's protocol, and the cDNA was amplified with the SYBR Green

Real-Time PCR assay kit (QIAGEN, Hilden, Germany). The transcript levels of *cpcA*, *apcA*, *cpcG1*, *cpcG2*, *apcE*, *apcF*, and *ndhI* (internal standard) were measured by RT-qPCR in triplicates using the 7500 Real-Time PCR system (Applied Biosystems, Foster City, CA, USA) using the primers shown in Supplementary Table 1.

**Construction of mutants**. The mutant strains of the top five upregulated unknown genes in our transcriptomic data were constructed as follows. The upstream and downstream regions of *ssr1698*, *sll0253*, *ssl0483*, *sll1654*, or *ssr1528* were amplified by PCR, creating appropriate restriction sites. A DNA fragment encoding a Kam$^R$ cassette was also amplified by PCR, creating the *Kpn*I and *Sac*I sites using appropriate PCR primers (Supplementary Table 1). These three PCR products were ligated into the MCS of pUC19 (Supplementary Fig. 5b) and used to transform the wild-type cells of *Synechocystis* 6803 to generate the Δ*ssr1698*, Δ*sll0253*, Δ*ssl0483*, Δ*sll1654* or Δ*ssr1528* mutant, respectively. Subsequently, these transformants were spread on agar plates containing BG-11 medium and kanamycin (10 μg mL$^{-1}$) buffered at pH 8.0, and the plates were incubated in 2% (*v/v*) $CO_2$ in air under illumination by fluorescent lamps at 40 μmol photons m$^{-2}$ s$^{-1}$. The mutated *ssr1698*, *sll0253*, *ssl0483*, *sll1654*, or *ssr1528* in the transformants was segregated to homogeneity (by successive streak purification) as determined by PCR amplification (Supplementary Fig. 5c).

The Δ*argZ* and Δ*ssr1698/argZ* mutants were constructed as follows. The upstream and downstream regions of *argZ* were amplified by PCR, creating appropriate restriction sites. A DNA fragment encoding a Sp$^R$ cassette was also amplified by PCR, creating the *Kpn*I and *Sac*I sites using appropriate PCR primers (Supplementary Table 1). They were ligated into the MCS of pUC19 to generate the pUC19-Δ*argZ* plastid. The pUC19-Δ*argZ* plastid was transformed into wild-type *Synechocystis* 6803 and its mutant Δ*ssr1698* to obtain Δ*argZ* and Δ*ssr1698/argZ*, respectively.

**Isolation of crude thylakoid membranes**. The cell cultures (800 mL) were harvested at different time points that the nitrate was added to chlorotic cells of wild-type *Synechocystis* 6803 or not, and the thylakoid membranes were isolated according to ref. [43] with some modifications as follows. The cells suspended in 5 mL of disruption buffer (10 mM HEPES-NaOH, 5 mM sodium phosphate, pH 7.5, 10 mM MgCl$_2$, 10 mM NaCl, and 25% [*v/v*] glycerol) were supplemented by zirconia/silica beads and broken by vortexing 15 times at the highest speed for 20 s at 4 °C with 5 min cooling on ice between the runs. The crude extract was centrifuged at 5000 × *g* for 5 min to remove the glass beads and unbroken cells. By further centrifugation at 20,000 × *g* for 30 min, we obtained crude thylakoid membranes from the precipitation.

**Electrophoresis and immunoblotting**. Blue native-PAGE (BN-PAGE) of *Synechocystis* 6803 membranes was performed as described previously[44] with slight modifications[45–52]. Isolated membranes were prepared for BN-PAGE as follows. Membranes were washed with 330 mM sorbitol, 50 mM Bis-Tris, pH 7, and 0.5 mM phenylmethylsulfonyl fluoride (Sigma-Aldrich, St. Louis, MO, USA) and resuspended in 20% (*w/v*) glycerol, 25 mM Bis-Tris, pH 7, 10 mM MgCl$_2$, 0.1 units of RNase-free DNase RQ1 (Promega, Madison, WI, USA) at a chlorophyll *a* concentration corresponding to 2 × 10$^9$ cells, and 0.5 mM phenylmethylsulfonyl fluoride. The samples were incubated on ice for 10 min, and an equal volume of 3% *n*-dodecyl β-D-maltoside (DM) was added. Solubilization was performed for 40 min on ice. Insoluble components were removed by centrifugation at 18,000 × *g* for 15 min. The collected supernatant was mixed with a one-tenth volume of

sample buffer, 5% Serva Blue G, 100 mM Bis-Tris, pH 7, 30% (*w/v*) Sucrose, 500 mM ε-amino-*n*-caproic acid, and 10 mM EDTA. Solubilized membranes were then applied to a 0.75-mm-thick, 5% to 12.5% acrylamide gradient gel (Hoefer Mighty Small mini-vertical unit, Hoefer, San Francisco, CA, USA). Samples were loaded on an equal cell number basis per lane. Electrophoresis was performed at 4 °C by increasing the voltage gradually from 50 up to 200 V during the 5.5-h run.

Clear native-PAGE (CN-PAGE) was used to detect the oligomerization of the purified tag-less Ssr1698 and was performed with 0.01% DM and 0.025% deoxycholate additives to the cathode buffer as described previously[53] with slight modifications[54,55]. The purified tag-less Ssr1698 were applied to a 0.75-mm-thick, 5–12.5% acrylamide gradient gel (Hoefer). Electrophoresis was performed at 4 °C by increasing the voltage gradually from 50 up to 200 V during the 5.5-h run.

SDS-PAGE of *Synechocystis* 6803 total proteins was performed on 12% polyacrylamide gel with 6 M urea as described previously[56]. After electrophoresis, the proteins were visualized by Coomassie Brilliant Blue (CBB) staining.

For immunoblotting, the proteins were electrotransferred to a polyvinylidene difluoride membrane (Immobilon-P; Millipore, Bedford, MA, USA) and detected by protein-specific antibodies using an ECL assay kit (Amersham, Arlington Heights, IL, USA) according to the manufacturer's protocol. Antibodies against CpcA (1:1000), ApcA (1:000), PsbO (1:1000), PsaD (1:1000), and NdhI (1:2000) were raised previously in our laboratory[48,57]. Antibodies against PsaB (1:5000) and D1 (1:5000) were purchased from Agrisera (Vännäs, Sweden).

**Phycobilisome isolation**. The phycobilisome was isolated according to the method of Glazer[58] with some modifications. The wild-type *Synechocystis* 6803 and its mutant Δ*ssr1698* cells were harvested at different time points that the nitrate was added to chlorotic cells or not, and washed twice in 0.75 M potassium phosphate buffer (pH 7.6). Subsequently, cells were resuspended in the same buffer and broken at 4 °C by a high-pressure homogenizer (PhD Technology LLC, Saint Paul, MN, USA) for three cycles at a pressure of 5000 psi. The crude extract was centrifuged at 5000 × *g* for 5 min at 4 °C to remove the unbroken cells and debris. Triton X-100 was added to the suspension to a final concentration of 2% (*v/v*) to release the phycobilisome from the thylakoid membranes. After 30 min of incubation (4 °C) at constant shaking, the thylakoids and debris were removed by centrifugation (20,000 × *g*, 30 min). The supernatant was loaded onto a linear (5–30%) sucrose density gradient in 0.75 M potassium phosphate buffer (pH 7.6) for ultracentrifugation at 110,000 × *g* for 16 h in a P40ST rotor (Hitachi, Tokyo, Japan). A linear sucrose gradient was prepared from 5.2 mL of 5% and 5.2 mL of 30% sucrose solutions in 0.75 M K-phosphate buffer using a gradient mixer (Hoefer). After the ultracentrifugation, the gathered narrow blue-colored zone was collected for further analysis.

**Expression and purification of fusion proteins**. The fragment containing *ssr1698* gene was amplified by PCR and inserted between the *Bam*HI and *Xho*I sites of pGEX-5X-1 to form the GST-tagged fusion protein construct (primers are shown in Supplementary Table 1). Subsequently, the construct was transformed into *E. coli* strain BL21 (DE3) pLysS and induced by 0.2 mM isopropyl-β-D-thiogalactoside for 12 h at 16 °C to express GST-Ssr1698 fusion protein. The fusion protein was purified at 4 °C using a GST column (GE Healthcare, Piscataway, NJ, USA), according to the manufacturer's instructions. Then size-exclusion chromatography with the Superdex 200 Increase 10/300GL

column (GE Healthcare) equilibrated with PBS buffer (pH 7.4) was used for further purification and the peak fraction containing GST-Ssr1698 was collected and concentrated. Finally, the tag-less Ssr1698 was purified by size-exclusion chromatography equilibrated with PBS buffer (pH 7.4) after the GST-tag was removed by Factor Xa protease (NEB, Beverley, MA, USA).

**Heme titration assays**. The heme-binding ability of tag-less Ssr1698 was estimated by optical titrations in PBS buffer (pH 7.4) as described previously[27,28] using ultraviolet–visible spectrophotometer (UV3000, Shimadzu, Kyoto, Japan) at room temperature. To exclude the possibility of nonspecific binding of heme, the binding reactions were conducted in the presence of 20 µM BSA. For the c-type heme-binding assay, MP-11 in 1 µM increments was added to both the sample and the reference cuvettes, the former of which contained 10 µM tag-less Ssr1698 and 20 µM BSA, and the latter of which contained 20 µM BSA and not tag-less Ssr1698. For the b-type heme-binding assay, hemin in 2 µM increments was added to both the sample and the reference cuvettes, the former of which contained 20 µM tag-less Ssr1698 and 20 µM BSA, and the latter of which contained 20 µM BSA and not tag-less Ssr1698. The difference absorption spectra were obtained by subtracting the spectrum of the reference cuvette from that of sample cuvette.

**Heme oxygenase activity assays**. The c-type heme oxygenase activity of Ssr1698 was examined by incubating either 10.8 µM Ssr1698 (enzyme) and 5.4 µM MP-11 (a substrate for c-type heme oxygenase) or 20 µM Ssr1698 (enzyme) and 10 µM hemin (a substrate for b-type heme oxygenase) with 10 mM ascorbic acid and 2 µM catalase in a cuvette containing PBS buffer (pH 7.4). The reaction was initiated by addition of sodium ascorbate and UV–visible absorption spectra was recorded every 30 min for 2 h using a UV3000 spectrophotometer (Shimadzu).

Products of reaction catalyzed by Ssr1698 were analyzed as follows. (1) Biliverdin IXα was detected by the liquid chromatography-mass spectrometry (LC-MS) using Agilent 1290UPLC/6550Q-TOF. 5 µL of the reaction products was applied to the ACQUITY UPLC BEH C18 column (2.1 × 100 mm, 1.7 µm, Waters, Milford, MA, USA) with a 300 µL/min flow rate of mobile phase solvent A (0.1% formic acid) and solvent B (methanol) using the following linear gradient: 5–40% solvent B for 3 min, 40–95% solvent B for 10 min, 95% solvent B for 4 min, 95–5% solvent B for 1 min, and 5% solvent B for 3 min. The LC-MS was recorded over a range of m/z 100–2000 units. The following parameters were used: the capillary voltage 4.0 kV for positive-ion mode; desolvation temperature 350 °C; desolvation gas 12 L/min. (2) Peptide was detected by Shanghai Luming Biological Technology Co. Ltd (Shanghai, China) using Q-Exactive mass spectrometer. The scanning range of parent ions was corrected by standard correction fluid at 300–1600 m/z, and the scanning mode of mass spectrometry was data-dependent acquisition (DDA). The 20 strongest fragment profiles (MS2 Scan) are collected after each full scan. Fragmentation was performed using high-energy collision dissociation (HCD, high energy) with an NCE energy of 28 and dynamic removal time of 25 s. The resolution of MS1 was 70000 at m/z 200, the AGC target was 3E6, and the maximum injection time was 100 ms. The resolution of MS2 was 17500, the AGC target was 1E5, and the maximum injection time was 50 ms. (3) Detection of free iron was performed as described previously[23,59]. 2.8 µM Ssr1698 was incubated with 10 mM ascorbic acid, 0–6 µM MP-11, 2 µM catalase, and 40 µL ferrozine reagent B (2 M ascorbic acid, 5 M ammonium acetate, 6.5 mM ferrozine, 13.1 mM neocuproine) in a cuvette containing a total of 400 µL of PBS buffer (pH 7.4). A control reaction was also performed without Ssr1698. The reactions were monitored at 562 nm using ultraviolet–visible spectrophotometer (Shimadzu).

The dependence of the initial rate of the c-type heme oxygenase reaction on the concentration of enzyme was determined by incubating 9 µM MP-11, 10 mM sodium ascorbate, 2 µM catalase, and 0–5 µM Ssr1698 in a cuvette containing PBS buffer (pH 7.4). The reactions were monitored at 406 nm and were performed in triplicate using Shimadzu UV3000 UV–visible spectrophotometer. The reaction velocity was calculated as the MP-11 consumption per minute.

**Chlorophyll fluorescence and P700 analysis**. Synechocystis 6803 cells were harvested at different time points that the nitrate was added to chlorotic cells or not, and the activities of their photosystem II (PSII) and PSI reaction centers were estimated by the $F_v/F_m$ and $P_m$ parameters, respectively. These two parameters were measured as follows.

The yield of chlorophyll (Chl) fluorescence at the steady state of electron transport was measured using a Dual-PAM-100 monitoring system (Walz, Effeltrich, Germany) equipped with an ED-101US/MD unit[60,61]. Minimal fluorescence corresponding to open PSII centers in the dark-adapted state ($F_o$) was excited by a weak measuring light (650 nm) at a PFD of 0.05–0.15 µmol photons m$^{-2}$ s$^{-1}$. A saturation pulse of red light (100 ms; 10,000 µmol photons m$^{-2}$ s$^{-1}$) was applied to determine the maximal fluorescence at closed PSII centers in the dark-adapted state ($F_m$), as described previously[62]. $F_v/F_m$ was calculated as $(F_m - F_o)/F_m$[63].

With the Dual-PAM-100, P700$^+$ was monitored as the absorption difference between 830 and 875 nm in transmission mode. The quantum yields of PSI were determined using the saturation pulse method as described previously[64,65]. The level of maximal P700$^+$ signal observed upon full oxidation, $P_m$, was determined by the application of a saturation pulse of red light (100 ms; 10,000 µmol photons m$^{-2}$ s$^{-1}$) in the presence of far-red light (about 0.3 µmol photons m$^{-2}$ s$^{-1}$) at 720 nm.

**77K fluorescence emission spectra**. Fluorescence emission spectra at 77 K of phycobilisome isolated from the wild-type Synechocystis 6803 and its mutant Δssr1698 that the nitrate was added to chlorotic cells or not for different time points were measured using a F4500 spectrofluorimeter (Hitachi) at an excitation wavelength of 580 nm. Fifty nanograms of isolated phycobilisome samples was rapidly frozen in liquid nitrogen. The fluorescence spectrum was obtained by averaging six scans for each sample in different tubes. The excitation and emission slit widths were set at 5 nm, and the same capture was used in all experiments. The phycobilisome activity was assessed by the ratio of fluorescence emission intensity at 683 nm (from the core component of phycobilisome) to that at 654 nm (from the rod component of phycobilisome)[66].

**Dry weight**. Dry weight (DW) of wild-type Synechocystis 6803 and its mutants during nitrogen oscillation or nitrogen repletion was measured according to the method of ref. [67] with some modifications. First of all, the cell cultures (10 mL) were harvested and washed twice with deionized water. Second, cells were resuspended in 100 µL deionized water and dropped on a pre-weighed polyvinylidene difluoride (PVDF) membrane (0.45 µm; Immobilon-P; Millipore, Bedford, MA, USA). Finally, the cell-bound PVDF membrane was evaporated at 65 °C for about 24 h, and the oven-dried membrane was weighed. DW was calculated according to the formula: DW (g L$^{-1}$ culture) = (oven-dried cell-bound PVDF membrane – PVDF membrane) × 100.

**Homology modeling of protein structure and molecular docking**. The protein structure of Ssr1698 was modeled by AlphaFold[68]. The structure of Ssr1698 dimer was predicted by ClusPro server[69]. Docking was performed using AutoDockVina[70]. The structure of dimeric Ssr1698 in complex with *c*-type heme was visualized using ChimeraX[71] and PyMOL[72].

**Homolog sequence of Ssr1698 in Tara Oceans Database**. The Ssr1698 protein sequence was queried against the Ocean Gene Atlas webserver (http://tara-oceans.mio.osupytheas.fr/ocean-gene-atlas/) using the Tara Oceans Microbiome Reference Gene Catalog version 1 (OM-RGC.v1) to search for cyanobacterial homologs[31] via clicking the size fractionation range of cyanobacterial cells[73]. Threshold *E* values are given in the main text.

**Statistics and reproducibility**. A minimum of three independent experiments were conducted for all studies, with specific replicate numbers (*n*) provided in the figure legends. Statistical analyses were carried out using Windows Excel. Significance between two groups was assessed using a two-tailed Student's *t* test and a *P* value less than 0.05 was considered statistically significant. The data were presented as the mean ± standard deviation.

**Reporting summary**. Further information on research design is available in the Nature Portfolio Reporting Summary linked to this article.

## Data availability

Transcriptome sequencing raw data have been deposited and are available in Gene Expression Omnibus under accession number GSE241350. All data generated or analyzed during this study have been included in both the published article and its supplementary information files. The numerical source data for all graphs are presented in the Supplementary Data 1. In addition, original uncropped blot/gel images, including the original size marker, have been made available in Supplementary Fig. 21. For more information, feel free to reach out to the corresponding authors upon a reasonable request.

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

## Acknowledgements

This work was supported by grants from the National Natural Science Foundation of China (32170257 and 31770259 to W.M., 32070381 to L.W.), Shanghai Science and Technology Committee (22010503500 to W.M., 20ZR1441300 to L.W.), Project of Young Scientist Exchange for one Belt and One Road Strategy in the International Science and Technology Cooperation of Shanghai Science and Technology Commission (19230742600 to D.L.), and Open Research Project for the Technology Innovation Center for Land Spatial Eco-restoration in Metropolitan Area, Ministry of Natural Resources (CXZX2021B01 and CXZX2022B01 to D.L.).

## Author contributions

W.M., Z.R., D.L., and L.W. initiated and orchestrated the project. Z.R. carried out transcriptomic analysis, physiological and biochemical analysis. Z.D. and M.Z. constructed and identified the mutants. G.M. analyzed the RT-qPCR data. L.L. made the liquid chromatography-mass spectrometry analysis. All authors contributed to data interpretation and the writing of the manuscript. W.M. and Z.R. wrote the manuscript.

## Competing interests

The authors declare no competing interests.
