## [Peer Review File · Communications Biology]

Reviewers' comments:

Reviewer #1 (Remarks to the Author):

This manuscript proposed that Ssr1698, expressed during recovery from nitrogen chlorosis, function as c-type heme oxygenase (cHO). This finding is of great interest if this is true because there have been no reports on cHO. However, the lack of experimental data raises the question of validity of this manuscript.

1. Construction of Ssr1698

1-1. The molecular mass of purified Ssr1698 is about 31 kDa based on the SDS-PAGE gel in Supplementary Fig. 6. This is a very large value as a 96-amino acid protein. Because the authors used pET32a for expression, the purified Ssr1698 is a fusion protein with thioredoxin (TrxA). The effect of TrxA on heme binding and heme oxygenase activity cannot be ignored. This fusion protein can be removed using thrombin or enterokinase. Removal of the Tag protein is necessary.

What about the oligomerization state of the purified Ssr1698? It can be estimated by the chromatogram of Supplementary Fig. 6.

1-2. Another problem is the presence of His-tag. pET32 contains His-tags at both N- and C-termini. Was Ssr1698 expressed with two His-tags or one? The His-tag also has the ability to interact with heme.

2. Absorption spectra of heme-bound Ssr1698

2-1. The absorption spectrum of Ssr1698 with heme in vitro (Fig. 3c) is different from that in vivo (Fig. 3b). Why are the spectra so different? According to the heme-binding ability assays, 2.8 μM apo-Ssr1698 was reconstituted with 25 μM hemin. Why does Ssr1698 require such an excess amount of hemin? Nearly 90% of hemin exists in the unbound form. Why does not the spectra of unbound hemin appear in Fig. 3c?

2-2. The ratio of the Soret band to absorbance at 280 nm contains information about the heme binding. The spectra of Figs. 3b, c, and Fig. 4 should contain the region from 250 to 800 nm.

3. Heme oxygenase activity

3-1. According to the heme oxygenase activity assays, almost 10 equivalents of hemin were added. What happens if the reaction is conducted under a single-turnover condition?

3-2. Please show the absorption spectra of Ssr1698 complexed with MP-11.

3-3. Ascorbic acid, which was used for measurement of the heme oxygenase activity of Ssr1698, can reduce molecular oxygen to produce hydrogen peroxide (H_2O_2). Therefore, the reaction of b-type HO is usually observed in the presence of catalase. The authors need to measure the reaction in the presence of catalase to scavenge H_2O_2 .

3-4 The reaction of mammalian HO with H_2O_2 produces verdoheme. What happens if Ssr1698/hemin or Ssr1698/MP-1 reacts with H_2O_2 ?

4. The most confusing point in this manuscript is the cleavage of the thioether bonds. Thioether bond is not normally cleaved by the reaction with ascorbic acid or H₂O₂. To prepare apocytochrome c to remove heme c, cytochrome c is treated with AgSO₄ in the presence of acetic acid (doi.org/10.1016/S0021-9258(19)44026-X). Thus, the reaction scheme shown in Fig. 4 is unacceptable. Please confirm the peptide from which heme is detached using MS spectrum.

Reviewer #2 (Remarks to the Author):

"Identification of a c-type heme oxygenase and its function during acclimation of cyanobacteria to nitrogen fluctuations" describes the discovery and characterization of a previously unknown protein, Ssr1698. The authors report that Ssr1698 is a c-type heme oxygenase that has a critical role on the phycobiliprotein biosynthesis pathway of *Synechocystis* 6803. This pathway is shown to be vital for the recovery of this organism from nitrogen chlorosis. Unfortunately, based upon the data presented in this manuscript, I am NOT convinced that Ssr1698 is a heme oxygenase. The authors need to do some additional experiments before reaching this conclusion.

Major Concerns

A. Although technically accurate, the Introduction of this manuscript is misleading. The manuscript leaves the reader with the impression that this is the first reported c-type heme oxygenase, which is not true. This may be the first c-type heme oxygenase found in cyanobacteria, but it is certainly not the first c-type heme oxygenase found in nature. The authors should briefly review the relevant c-type heme oxygenase literature in the Introduction to provide the reader with relevant background information and clarify the novelty of this work.

B. It appears that all of the heme binding and activity assays were carried out with His-tagged protein, which is problematic. In general, His-tags are problematic for bioinorganic proteins because His can ligate metals. For heme oxygenases specifically, His tags are problematic because His is a common heme ligand. The heme could be simply ligating to the His-tag and degrading via coupled oxidation. The authors should repeat the heme binding and activity assays for protein constructs lacking His tags.

C. How are the thioether linkages cleaved by Ssr1698? Heme oxygenases cleave the porphyrin ring yielding iron, biliverdin, and CO products but they are not known to cleave the thioether linkages that tether c-type heme to the polypeptide. The data in Figure 4 clearly shows that the enzyme produces a biliverdin product, and the reaction scheme within Figure 4 describes the overall reaction, but it does not explain how this enzyme cleaves the two Cys-porphyrin bonds.

D. Figure 3B/C. First, why is there a 10 nm shift between the UV/Vis Abs spectra of

"Purified Ssr1698 with heme" in panel B and Heme-Ssr1698 in panel C? A 10 nm shift of the heme Soret band is extremely significant; these are two different species. Second, the UV/Vis Abs spectrum of heme is a relatively good fingerprint of the heme axial ligation. Are either of the UV/Vis Abs spectra shown in Figure 3 consistent with the binding site predicted in panel D?

E. Figure 3D. The authors depict the docking of heme b to Ssr1698 in Figure 3D based upon AlphaFold and COFACTOR, but heme b is not the substrate of this enzyme! The substrate is MP-11, which is much larger and depicted in Figure 4.

F. Figure 4A. How does the "0 min" UV/Vis absorption trace compare with the UV/Vis absorption spectrum of free MP-11? Axial ligand changes, as might be expected for heme binding by Ssr1698, trigger significant spectroscopic changes. You should be able to assess whether Ssr1698 is binding heme-c, and perhaps identify the axial ligands, based upon the UV/Vis absorption data presented in panel A.

Reviewer #3 (Remarks to the Author):

In the manuscript by Ran et al., the authors identified several genes in the cyanobacterium *Synechocystis* sp. strain PCC 6803 that are upregulated during recovery from nitrogen-deficiency induced chlorosis via nitrate exposure. One of these genes, *ssr1698* encodes a protein that is homologous to the N-terminal DUF2470 domain of the non-canonical heme oxygenase HugZ. A knockout strain of this gene showed a slight decrease in viability upon recovery from nitrogen chlorosis and diminished production of phycobiliproteins (which the authors showed accumulates prior to that of photosystem proteins). Based on these observations, the authors speculated that Ssr1698 could play a role in the production of phycobilins (e.g., via heme oxygenase activity). To test this, the authors showed that polyhistidine-tagged Ssr1698 can bind heme, although no heme oxygenase activity was observed with b-type heme. In contrast, some heme degradation activity was observed with a c-type hemopeptide. From this data, the authors conclude that Ssr1698 is a novel c-type heme oxygenase that produces the phycobilin precursor biliverdin IX α . However, there are several issues with these data/conclusions that dampen enthusiasm for this manuscript.

First, Ssr1698 was purified with a polyhistidine-tag. The heme binding-properties of the HugZ homolog HupZ was shown by Traore, et al. (*Molecules* 26, 2021, 549) to be related to the presence of the tag.

Additionally, the "rapid" c-type heme degradation activity that was observed was ~40% substrate consumption over 2 hours. The activity of the first bona fide c-type heme oxygenase, Pden1313, showed 100% conversion in 5 min (*J. Bio. Chem.* 296, 2021, 100666). It is possible that the polyhistidine-tag could be interfering with the activity of Ssr1698. However, both b- and c-type heme can spontaneously convert to the corresponding biliverdin isomers. The fact that no spontaneous oxidation was observed with b-type heme could be due to a protective effect of free heme binding to

Ssr1698.

Finally, both HugZ and Pden1323 catalysis proceeds with cleavage at the δ - and/or β -meso positions rather than the α -meso position. Thus, the assignment of product of the reaction as biliverdin IX α is questionable without additional data.

For these reasons, it is not recommended that this manuscript be accepted in its present form.

April 13, 2023

Professor Haichun Gao
Editorial Board Member
Communications Biology

Dear Professor Gao,

Thank you for handling our manuscript “Identification of a *c*-type heme oxygenase and its function during acclimation of cyanobacteria to nitrogen fluctuations” (COMMSBIO-22-3505A).

We appreciate reviewers’ constructive comments and have added a lot of new experiments (Figs. 3 and 4; Extended Data Figs. 3 and 4; Supplementary Figs. 6-9) in the revised manuscript to strengthen our conclusion that Ssr1698 functions as a *c*-type heme oxygenase in cyanobacteria. In addition, we have added the fact in the revised “Introduction” section that such *c*-type heme oxygenase has been previously reported in bacteria (page 4, lines 64 to 72). Here we submit our revision of the paper with a point-to-point response to the reviewers’ comments.

Thank you very much for your help. We are looking forward to hearing from you about the revised manuscript.

Best regards,

Weimin Ma, PhD, Professor
College of Life Sciences
Shanghai Normal University
Shanghai 200234, China
Tel: 86-21-6432-1617 (Office)
E-mail: wma@shnu.edu.cn

Point-to-point response to reviewers' comments

Reviewer #1

This manuscript proposed that Ssr1698, expressed during recovery from nitrogen chlorosis, function as c-type heme oxygenase (cHO). This finding is of great interest if this is true because there have been no reports on cHO. However, the lack of experimental data raises the question of validity of this manuscript.

Thanks! We have added a lot of new experiments (Figs. 3 and 4; Extended Data Figs. 3 and 4; Supplementary Figs. 6-9) in the revised manuscript to strengthen our conclusion that Ssr1698 functions as a c-type heme oxygenase.

1. Construction of Ssr1698

1-1. The molecular mass of purified Ssr1698 is about 31 kDa based on the SDS-PAGE gel in Supplementary Fig. 6. This is a very large value as a 96-amino acid protein. Because the authors used pET32a for expression, the purified Ssr1698 is a fusion protein with thioredoxin (TrxA). The effect of TrxA on heme binding and heme oxygenase activity cannot be ignored. This fusion protein can be removed using thrombin or enterokinase. Removal of the Tag protein is necessary. What about the oligomerization state of the purified Ssr1698? It can be estimated by the chromatogram of Supplementary Fig. 6.

Thanks for this good point. Following this reviewer's suggestion, we constructed the pGEX-*ssr1698* plasmid and purified the tag-less Ssr1698 protein after GST tag was removed by Factor Xa protease (see updated Supplementary Fig. 6). Subsequently, the purified tag-less Ssr1698 protein was used to remeasure the heme binding and heme oxygenase activity (see updated Figs. 3 and 4; Extended Data Figs. 3 and 4). In addition, the dimeric state of the purified tag-less Ssr1698 is supported by the data of size-exclusion chromatogram (see updated Supplementary Fig. 6), and CN-PAGE (see new Fig. 3d).

(Page 18, lines 359 to 363), "In addition, the ratio of heme to Ssr1698 suggested that two Ssr1698 proteins as a dimer share a heme molecule (Fig. 3c; Extended Data Fig. 3), as shown in the predicted structure (Fig. 3e). This dimeric state of Ssr1698 was supported by the data of size-exclusion chromatogram (Supplementary Fig. 6) and CN-PAGE (Fig. 3d)."

1-2. Another problem is the presence of His-tag. pET32 contains His-tags at both N- and C-termini. Was Ssr1698 expressed with two His-tags or one? The His-tag also has the ability to interact with heme.

Thanks for this good point. Following this reviewer's suggestion, we constructed the pGEX-*ssr1698* plasmid and purified the tag-less Ssr1698 protein after GST tag was

removed by Factor Xa protease (see updated Supplementary Fig. 6). Subsequently, the purified tag-less Ssr1698 protein was used to remeasure the heme binding and heme oxygenase activity (see updated Figs. 3 and 4; Extended Data Figs. 3 and 4).

2. Absorption spectra of heme-bound Ssr1698

2-1. The absorption spectrum of Ssr1698 with heme *in vitro* (Fig. 3c) is different from that *in vivo* (Fig. 3b). Why are the spectra so different? According to the heme-binding ability assays, 2.8 μM apo-Ssr1698 was reconstituted with 25 μM hemin. Why does Ssr1698 require such an excess amount of hemin? Nearly 90% of hemin exists in the unbound form. Why does not the spectra of unbound hemin appear in Fig. 3c?

Thanks for this good point. Such differences of Soret band may be caused by a large number of unbound free heme. Considering the potential effect of tag and unbound heme on the absorption spectra of Ssr1698 in complex with heme, we re-performed the absorption spectra of 5.4 μM MP-11 (*c*-type heme) with 0 to 21.6 μM purified tag-less Ssr1698 (see updated Fig. 3) or 10 μM hemin (*b*-type heme) with 0 to 40 μM purified tag-less Ssr1698 (see updated Extended Data Fig. 3).

(Page 11, line 221 to page 12, line 230), “The heme-binding ability of tag-less Ssr1698 was determined by spectroscopic method. Absorption spectra of the purified tag-less Ssr1698 and heme mixture were monitored on ultraviolet-visible spectrophotometer (UV3000, Shimadzu, Kyoto, Japan) between 250 and 800 nm at room temperature. For the *c*-type heme-binding assay, the absorption spectra were monitored after the 5.4 μM MP-11 was incubated with different concentrations of tag-less Ssr1698 (0, 2.7, 5.4, 8.1, 10.8, 13.5, 16.2, 18.9, and 21.6 μM) for 5 min at room temperature. For the *b*-type heme-binding assay, the absorption spectra were monitored after the 10 μM hemin was incubated with different concentrations of tag-less Ssr1698 (0, 5, 10, 15, 20, 25, 30, 35 and 40 μM) for 30 min at room temperature.”

2-2. The ratio of the Soret band to absorbance at 280 nm contains information about the heme binding. The spectra of Figs. 3b, c, and Fig. 4 should contain the region from 250 to 800 nm.

Thanks for this good point. Following this reviewer’s suggestion, the region from 250 to 800 nm has been displayed in the absorbance spectra of Figs. 3 and 4; Extended Data Figs. 3 and 4; and Supplementary Fig. 9 in the revised manuscript.

3. Heme oxygenase activity

3-1. According to the heme oxygenase activity assays, almost 10 equivalents of hemin were added. What happens if the reaction is conducted under a single-turnover condition?

Thanks for this good point. Following this reviewer's suggestion, we have re-performed the heme oxygenase activity assays under a single-turnover condition (see updated Fig. 4a and Extended Data Fig. 4b).

3-2. Please show the absorption spectra of Ssr1698 complexed with MP-11.

Following this reviewer's suggestion, we have added the absorption spectra of purified tag-less Ssr1698 complexed with MP-11 in the revised manuscript (see updated Fig. 3b).

3-3. Ascorbic acid, which was used for measurement of the heme oxygenase activity of Ssr1698, can reduce molecular oxygen to produce hydrogen peroxide (H₂O₂). Therefore, the reaction of b-type HO is usually observed in the presence of catalase. The authors need to measure the reaction in the presence of catalase to scavenge H₂O₂.

We thank the reviewer for this comment. Following this reviewer's suggestion, we have added the enzyme catalase to the ascorbic acid reaction system (page 12, lines 232 to 238) and have re-performed the heme oxygenase activity assays (see updated Fig. 4a; and Extended Data Fig. 4b).

(Page 11, lines 231 to 237), "The *c*-type heme oxygenase activity of Ssr1698 was examined by incubating either 10.8 μM Ssr1698 (enzyme) and 5.4 μM MP-11 (a substrate for *c*-type heme oxygenase) or 20 μM Ssr1698 (enzyme) and 10 μM hemin (a substrate for *b*-type heme oxygenase) with 10 mM ascorbic acid and 2 μM catalase in a cuvette containing PBS buffer (pH 7.4). The reaction was initiated by addition of sodium ascorbate and UV-visible absorption spectra was recorded every 30 min for 2 h using a UV3000 spectrophotometer (Shimadzu)."

3-4 The reaction of mammalian HO with H₂O₂ produces verdoheme. What happens if Ssr1698/hemin or Ssr1698/MP-1 reacts with H₂O₂?

Following this reviewer's suggestion, we tried to test whether the reaction system can produce verdoheme in the presence of H₂O₂. It has been previously reported that the verdoheme production is closely associated with an increase in visible absorption in 660-690 nm region and a decrease in the Soret band (Liu et al., 1997; Ratliff et al., 2001). In addition, the decrease in the Soret band can be caused by the spontaneous reaction of MP-11 or hemin with H₂O₂ (Kremer, 1989; Araujo et al., 2007). Collectively, the verdoheme production is indicated as the increase in visible absorption in 660-690 nm region.

Our data clearly indicated that the Soret band is considerably decreased in the H₂O₂ reaction system (Responded Data Fig. 1). However, the visible absorption in 660-690 nm region is hardly increased in the H₂O₂ reaction system regardless of only presence

of heme or simultaneous presence of purified tag-less Ssr1698 and heme (Responded Data Fig. 1). We thus conclude that Ssr1698 is not involved in producing verdoheme in the H₂O₂ reaction system.

Responded Data Fig. 1 | Ssr1698 catalyzes the reaction of heme driven by H₂O₂. **a, c** For the reaction of *b*-type heme catalyzed by Ssr1698, H₂O₂ was added to the reaction system containing only hemin (**a**) and hemin and Ssr1698 mixture (**c**). **b, d** For the reaction of *c*-type heme catalyzed by Ssr1698, H₂O₂ was added to the reaction system containing only hemin (**b**) and hemin and Ssr1698 mixture (**d**).

4. The most confusing point in this manuscript is the cleavage of the thioether bonds. Thioether bond is not normally cleaved by the reaction with ascorbic acid or H₂O₂. To prepare apocytchrome c to remove heme c, cytochrome c is treated with AgSO₄ in the presence of acetic acid (doi.org/10.1016/S0021-9258(19)44026-X). Thus, the reaction scheme shown in Fig. 4 is unacceptable. Please confirm the peptide from which heme is detached using MS spectrum.

We thank the reviewer for this comment. Following this review's suggestion, we have identified an 11-amino acid peptide in the reaction product catalyzed by the Ssr1698 enzyme with *c*-type heme as a substrate using Q-Exactive mass spectrometer (see new Fig. 4d). In addition, a motif contained in Ssr1698 is similar to the HXXEH motif, which has been reported to be closely associated with the cleavage of thioether bond (Pei and Zhu, 2004). Collectively, Ssr1698 as a *c*-type heme oxygenase can catalyze the cleavage of thioether bond (see page 23, lines 456 to 462).

(Page 23, lines 456 to 462), “In cyanobacteria, the photosynthetic *c*HO enzyme can catalyze the cleavage of thioether bond, since we identify an 11-amino acid peptide in the reaction product catalyzed by the enzyme with *c*-type heme as a substrate (Fig. 4d). The cleavage is supported by the presence of a conserved HXXXDH motif in photosynthetic *c*HO (Supplementary Fig. 7) because a similar HXXEH motif in *S*-ribosylhomocysteinase (LuxS) has been reported to be closely associated with the non-redox cleavage of thioether bond [71].”

Reviewer #2

"Identification of a *c*-type heme oxygenase and its function during acclimation of cyanobacteria to nitrogen fluctuations" describes the discovery and characterization of a previously unknown protein, Ssr1698. The authors report that Ssr1698 is a *c*-type heme oxygenase that has a critical role on the phycobiliprotein biosynthesis pathway of *Synechocystis* 6803. This pathway is shown to be vital for the recovery of this organism from nitrogen chlorosis. Unfortunately, based upon the data presented in this manuscript, I am NOT convinced that Ssr1698 is a heme oxygenase. The authors need to do some additional experiments before reaching this conclusion.

Thanks! We have added a lot of new experiments (Figs. 3 and 4; Extended Data Figs. 3 and 4; Supplementary Figs. 6-9) in the revised manuscript to strengthen our conclusion that Ssr1698 functions as a *c*-type heme oxygenase in cyanobacteria.

Major Concerns

A. Although technically accurate, the Introduction of this manuscript is misleading. The manuscript leaves the reader with the impression that this is the first reported *c*-type heme oxygenase, which is not true. This may be the first *c*-type heme oxygenase found in cyanobacteria, but it is certainly not the first *c*-type heme oxygenase found in nature. The authors should briefly review the relevant *c*-type heme oxygenase literature in the Introduction to provide the reader with relevant background information and clarify the novelty of this work.

Thanks for this good point. Following this reviewer’s suggestion, we have added the relevant background knowledge in the revised “Introduction” section (page 4, lines 68 to 70).

(Page 4, lines 68 to 70), “In cyanobacteria, two canonical types of *b*-type heme oxygenase, *b*HO-1 and *b*HO-2, have been identified [21, 22], but the *c*-type heme oxygenase (*c*HO) remains a mystery, although it has been identified in bacteria [23, 24].”

B. It appears that all of the heme binding and activity assays were carried out with His-tagged protein, which is problematic. In general, His-tags are problematic for

bioinorganic proteins because His can ligate metals. For heme oxygenases specifically, His tags are problematic because His is a common heme ligand. The heme could be simply ligating to the His-tag and degrading via coupled oxidation. The authors should repeat the heme binding and activity assays for protein constructs lacking His tags.

Thanks for this good point. Following this reviewer's suggestion, we have constructed the pGEX-*ssr1698* plasmid and have purified the tag-less Ssr1698 protein after GST tag was removed by Factor Xa protease (see updated Supplementary Fig. 6). Subsequently, the purified tag-less Ssr1698 protein was used to remeasure the heme binding and heme oxygenase activity (see updated Figs. 3 and 4; Extended Data Figs. 3 and 4). Please also refer to our response to question 1 of the reviewer 1.

C. How are the thioether linkages cleaved by Ssr1698? Heme oxygenases cleave the porphyrin ring yielding iron, biliverdin, and CO products but they are not known to cleave the thioether linkages that tether c-type heme to the polypeptide. The data in Figure 4 clearly shows that the enzyme produces a biliverdin product, and the reaction scheme within Figure 4 describes the overall reaction, but it does not explain how this enzyme cleaves the two Cys-porphyrin bonds.

We thank the reviewer for this comment. Following this reviewer's suggestion, we have identified an 11-amino acid peptide in the reaction product catalyzed by the Ssr1698 enzyme with c-type heme as a substrate using Q-Exactive mass spectrometer (see new Fig. 4d). In addition, we found that a motif contained in Ssr1698 is similar to the HXXEH motif in S-ribosylhomocysteinase (LuxS), which has been reported to be closely associated with the non-redox cleavage of thioether bond (Pei and Zhu, 2004). Please also refer to our response to question 4 of the reviewer 1.

(Page 23, lines 456 to 462), "In cyanobacteria, the photosynthetic cHO enzyme can catalyze the cleavage of thioether bond, since we identify an 11-amino acid peptide in the reaction product catalyzed by the enzyme with c-type heme as a substrate (Fig. 4d). The cleavage is supported by the presence of a conserved HXXXDH motif in cyanobacterial cHO (Supplementary Fig. 7) because a similar HXXEH motif in S-ribosylhomocysteinase (LuxS) has been reported to be closely associated with the non-redox cleavage of thioether bond [71]."

D. Figure 3B/C. First, why is there a 10 nm shift between the UV/Vis Abs spectra of "Purified Ssr1698 with heme" in panel B and Heme-Ssr1698 in panel C? A 10 nm shift of the heme Soret band is extremely significant; these are two different species. Second, the UV/Vis Abs spectrum of heme is a relatively good fingerprint of the heme axial ligation. Are either of the UV/Vis Abs spectra shown in Figure 3 consistent with the binding site predicted in panel D?

Thanks for this good point. Such differences of Soret band may be caused by a large

number of unbound free heme. Considering the potential effect of unbound free heme and tag on the absorption spectra of Ssr1698 in complex with heme, we re-performed the absorption spectra of 5.4 μM MP-11 (*c*-type heme) with 0 to 21.6 μM purified tag-less Ssr1698 (see updated Fig. 3) or 10 μM hemin (*b*-type heme) with 0 to 40 μM purified tag-less Ssr1698 (see updated Extended Data Fig. 3). Please also refer to our response to question 2 of the reviewer 1.

As could be judged from the α and β bands between 500 and 600 nm (see new Fig. 3b and Extended data 3a), an axial heme ligand exists in *c*-type heme complexed with dimeric Ssr1698 but not in *b*-type heme complexed with dimeric Ssr1698, consistent with the binding sites predicted in Fig. 3e and Extended data Fig. 3d.

E. Figure 3D. The authors depict the docking of heme b to Ssr1698 in Figure 3D based upon AlphaFold and COFACTOR, but heme b is not the substrate of this enzyme! The substrate is MP-11, which is much larger and depicted in Figure 4.

Thanks for this good point. Following this reviewer's suggestion, we have re-predicted the structure of dimeric Ssr1698 in complex with *c*-type heme based on AlphaFold, ClusPro server, and AutoDockVina (see new Fig. 3e).

(Page 16, lines 308 to 311), "The protein structure of Ssr1698 was modeled by AlphaFold [51]. The structure of Ssr1698 dimer was predicted by ClusPro server [52]. Docking was performed using AutoDockVina [53]. The structure of dimeric Ssr1698 in complex with *c*-type heme or *b*-type heme was visualized using ChimeraX [54] and PyMOL [55]."

F. Figure 4A. How does the "0 min" UV/Vis absorption trace compare with the UV/Vis absorption spectrum of free MP-11? Axial ligand changes, as might be expected for heme binding by Ssr1698, trigger significant spectroscopic changes. You should be able to assess whether Ssr1698 is binding heme-*c*, and perhaps identify the axial ligands, based upon the UV/Vis absorption data presented in panel A.

Thanks for this good point. Following this reviewer's suggestion, we have compared the UV/Vis absorption spectra of free MP-11 with MP-11 complexed with Ssr1698 at "0 min". Our data indicated their difference (see new Fig. 3b) and supported the binding of *c*-type heme with dimeric Ssr1698 (Fig. 3e).

Reviewer #3

In the manuscript by Ran et al., the authors identified several genes in the cyanobacterium *Synechocystis* sp. strain PCC 6803 that are upregulated during recovery from nitrogen-deficiency induced chlorosis via nitrate exposure. One of these genes, *ssr1698* encodes a protein that is homologous to the N-terminal

DUF2470 domain of the non-canonical heme oxygenase HugZ. A knockout strain of this gene showed a slight decrease in viability upon recovery from nitrogen chlorosis and diminished production of phycobiliproteins (which the authors showed accumulates prior to that of photosystem proteins). Based on these observations, the authors speculated that Ssr1698 could play a role in the production of phycobilins (e.g., via heme oxygenase activity). To test this, the authors showed that polyhistidine-tagged Ssr1698 can bind heme, although no heme oxygenase activity was observed with b-type heme. In contrast, some heme degradation activity was observed with a c-type hemopeptide. From this data, the authors conclude that Ssr1698 is a novel c-type heme oxygenase that produces the phycobilin precursor biliverdin IX α . However, there are several issues with these data/conclusions that dampen enthusiasm for this manuscript.

Thanks! We have added a lot of new experiments (Figs. 3 and 4; Extended Data Figs. 3 and 4; Supplementary Figs. 6-9) in the revised manuscript to strengthen our conclusion that Ssr1698 functions as a c-type heme oxygenase in cyanobacteria.

First, Ssr1698 was purified with a polyhistidine-tag. The heme binding-properties of the HugZ homolog HupZ was shown by Traore, et al. (Molecules 26, 2021, 549) to be related to the presence of the tag.

Thanks for this good point. Following this reviewer's suggestion, we have constructed the pGEX-*ssr1698* plasmid and have purified the tag-less Ssr1698 protein after GST tag was removed by Factor Xa protease (see updated Supplementary Fig. 6). Subsequently, the purified tag-less Ssr1698 protein was used to remeasure the heme binding and heme oxygenase activity (see updated Figs. 3 and 4; Extended Data Figs. 3 and 4). Please also refer to our response to question 1 of the reviewer 1 and question B of the reviewer 2.

Additionally, the "rapid" c-type heme degradation activity that was observed was ~40% substrate consumption over 2 hours. The activity of the first bona fide c-type heme oxygenase, Pden1313, showed 100% conversion in 5 min (J. Bio. Chem. 296, 2021, 100666). It is possible that the polyhistidine-tag could be interfering with the activity of Ssr1698. However, both b- and c-type heme can spontaneously convert to the corresponding biliverdin isomers. The fact that no spontaneous oxidation was observed with b-type heme could be due to a protective effect of free heme binding to Ssr1698.

Thanks for this good point. Following this reviewer's suggestion, we have remeasured the heme oxygenase activity of purified tag-less Ssr1698 protein (see updated Fig. 4; Extended Data Fig. 4). The results indicated that as expected by the reviewer, the polyhistidine-tag interferes with the activity of Ssr1698, although this activity is still relatively low compared with the previously reported c-type heme oxygenase in bacteria (Li et al., 2021). This difference may be caused by their different domains

that cyanobacterial *c*HO contains DUF2470 domain (Fig. 3a) and bacterial *c*HO contains β -barrel domain (Li et al., 2021).

Responded Data Fig. 2 | Ssr1698 catalyzes the reaction of *c*-type heme driven by ascorbic acid. a, b Spontaneous oxidation of free *b*-type heme (a) and its degradation is completely suppressed by free *b*-type heme binding to Ssr1698 (b). **c, d** Spontaneous oxidation of free *c*-type heme (c) and its degradation is accelerated by free *c*-type heme binding to Ssr1698 (d).

We found that the spontaneous oxidation of free *b*-type heme was completely suppressed by Ssr1698 (Responded Data Fig. 2a, b), indicating a protective effect of free *b*-type heme binding to Ssr1698. However, the degradation of free *c*-type heme was accelerated by free *c*-type heme binding to Ssr1698 (Responded Data Fig. 2c, d), strengthening our conclusion that Ssr1698 functions as a *c*-type heme oxygenase.

Finally, both HugZ and Pden1323 catalysis proceeds with cleavage at the δ - and/or β -meso positions rather than the α -meso position. Thus, the assignment of product of the reaction as biliverdin IX α is questionable without additional data. For these reasons, it is not recommended that this manuscript be accepted in its present form.

Thanks for this good point. The opening of the heme ring at the α -meso position is supported by the predicted structure that the polar amino acids of Ssr1698 dimer only observed close to the α -meso position of *c*-type heme may contribute to the formation of a hydrogen-bonding network near its α -meso position (Supplementary Fig. 8), which is required for the opening of the heme ring (Unno et al., 2007). The opening of

the heme ring at the α -meso position is also supported by the results that the degradation of *c*-type heme is significantly suppressed after the addition of biliverdin IX α to the reaction system catalyzed by the cyanobacterial *c*HO enzyme (Supplementary Fig. 9).

(Page 23, lines 462 to 470), “The enzyme can also catalyze the opening of the heme ring at the α -meso position to form biliverdin IX α (Fig. 4c). The opening is supported by the predicted structure that the polar amino acids of Ssr1698 dimer only observed close to the α -meso position of *c*-type heme may contribute to the formation of a hydrogen-bonding network near its α -meso position (Supplementary Fig. 8), which is required for the opening of the heme ring [72]. The opening is also supported by the results that the degradation of *c*-type heme is significantly suppressed after the addition of biliverdin IX α to the reaction system catalyzed by the cyanobacterial *c*HO enzyme ([73]; Supplementary Fig. 9).”

References

- Araujo JC, Prieto T, Prado FM, Trindade FJ, Nunes GL, dos Santos JG, et al. Peroxidase catalytic cycle of MCM-41-entrapped microperoxidase-11 as a mechanism for phenol oxidation. *J Nanosci Nanotechnol.* 2007; 7:3643–3652.
- Kremer ML. The reaction of hemin with H₂O₂. *Eur J Biochem.* 1989; 185:651–658.
- Li S, Isiorho EA, Owens VL, Donnan PH, Odili CL, Mansoorabadi SO. A noncanonical heme oxygenase specific for the degradation of *c*-type heme. *J Biol Chem.* 2021;296:100666.
- Liu Y, Moëne-Loccoz P, Loehr TM, De Montellano PRO. Heme oxygenase-1, intermediates in verdoheme formation and the requirement for reduction equivalents. *J Biol Chem.* 1997; 272:6909–6917.
- Pei, D, Zhu, J. Mechanism of action of S-ribosylhomocysteine (LuxS). *Curr Opin Chem Biol.* 2004; 8:492–497.
- Ratliff M., Zhu W., Deshmukh R., Wilks A., Stojiljkovic I. Homologues of neisserial heme oxygenase in gram-negative bacteria: degradation of heme by the product of the *pigA* gene of *Pseudomonas aeruginosa*. *J Bacteriol.* 2001; 183:6394–6403.
- Unno, M, Matsui, T, Ikeda-Saito, M. Structure and catalytic mechanism of heme oxygenase. *Nat Prod Rep.* 2007; 24:553–570.

Reviewers' comments:

Reviewer #1 (Remarks to the Author):

As the authors have revised appropriately the manuscript according to the reviewers' comments, I recommend to accept this paper for the publication in *Communications Biology*.

Reviewer #2 (Remarks to the Author):

The authors have addressed the majority of my comments, and I now support publication of this article in *Communications Biology*.

However, I have low confidence in the Alpha-Fold structure. The binding site of MP-11 in the revised manuscript is completely different from the binding site in the original manuscript. Ultimately, Alpha-fold is based upon homology modeling, and this is a novel protein with an unusual substrate MP-11. I don't think there is a reason to have high confidence in the Alpha-Fold structure. I suggest the authors considering removing this from the manuscript.

Reviewer #3 (Remarks to the Author):

Ran et al. made a number of revisions to their manuscript to address concerns regarding the characterization of a gene, *ssr1698*, that is upregulated during recovery from nitrogen-deficiency induced chlorosis via nitrate exposure. Concerns regarding artifactual effects due to the presence of a polyhistidine-tag on Ssr1698 were satisfactorily addressed. However, there are still many concerns that remain regarding the dual role of Ssr1698 as a novel c-heme oxygenase and phycobiliprotein maturase, and thus of its role in light-harvesting and nitrogen storage (i.e., the entire premise of the manuscript). These concerns are summarized below and should be strongly considered and/or addressed (either through experimentation or tempering of conclusions) if the manuscript is to be acceptable for publication.

It seems clear that the cell yield of *Synechocystis* sp. strain PCC 6803 increases upon nitrogen repletion and that deletion of *ssr1698* has a modest effect on the viability of this cyanobacterium in a light-dependent manner (e.g., Fig. 5d and Extended Data Fig. 5). However, there is contradictory data on whether repeated cycles of nitrogen depletion/repletion increases or decreases growth in response to the presence/absence of *ssr1698* (Fig. 5c vs. Extended Data Fig. 6).

Likewise, it is unclear whether the enhanced photosystem and phycobiliprotein production upon nitrogen repletion is merely a byproduct of increased cell yield or even has anything to do with the presence of *ssr1698*. Specifically, it is unclear if the data in Fig. 1 have been properly normalized to the cell dry weight at each time point (the protein gels from low and high nitrogen conditions in Extended Data Fig. 1 do

not appear to be). (Also, labels are missing for the traces in Fig. 1e.) Clearly, phycobiliproteins are still being produced in the Δ ssr1698 strain at comparable levels to wild-type (Fig. 2d and 2e), so this gene is not essential for, or the likely primary source of, phycobilin production.

Relatedly, it is not clear that phycobiliproteins are selectively used as a nitrogen reservoir. All proteins contain nitrogen and there is nothing in the data to suggest that phycobiliproteins are special in this regard over any other protein, save for their high abundance. Thus, it is odd at best and misleading at worst to highlight this feature, much less the role of ssr1698 in modulating these effects as a mechanism of the nitrogen starvation response of *Synechocystis* sp.

Similarly, the data presented in the revised manuscript for Ssr1698 as a novel c-heme oxygenase involved in phycobiliprotein maturation remains inconclusive. The UV-vis spectral changes taken as evidence for the 1:2 binding stoichiometry of both b- and c-heme to Ssr1698 is not convincing. The effects of [Ssr1698] on the absorption spectrum of heme are subtle and controls should be used to rule out solvation effects or non-specific binding (e.g., by titrating in glycerol/PEG or BSA). The fact that the heme spectrum is heme-dependent, as stated in the text (line 358), does not provide evidence that Ssr1698 binds heme.

It is curious/concerning that the predicted binding site of b- and c-heme are not the same (Fig. 3e and Extended Data Fig. 3d), and that neither site is consistent with the site to which heme binds in HupZ (the heme-binding site of the DUF2470 domain of HupZ is on the opposite side of that predicted for the Ssr1698 monomer). That the proposed binding site of c-heme has polar residues (lines 462-467 and Supplementary Fig. 8) provides no evidence for whether or not Ssr1698 cleaves c-heme with α -meso regiochemistry (e.g., the polarity of this region could simply be a reflection of its solvent accessibility). Such regiochemistry would be necessary if Ssr1698 indeed was involved in the biosynthesis of phycobilin, but the authors have provided no such confirmation of this fact. Additional experiments (e.g., mass spec fragmentation patterns or comparison with biliverdin isomeric standards) would be needed to confirm the regiospecificity of Ssr1698, assuming it is a heme oxygenase at all.

This leads to the last and most critical point that the observed c-heme oxygenase activity of Ssr1698 has not been conclusively shown to be enzymatic. The observed spectral changes and time-course of “Ssr1698-catalyzed” MP-11 degradation (Fig. 4a) are reminiscent of those of HupZ (Traore et al, 2021), wherein it was concluded that this enzyme was likely NOT a heme oxygenase. This could be confirmed with proper no enzyme controls (for both b- and c-heme) and showing that the rate of the observed heme degradation activity scales linearly with [Ssr1698]. Regardless, the “Ssr1698-catalyzed” reaction is certainly not rapid (as claimed on line 380 and in the legend of Fig. 4), particularly when compared to a bona fide c-heme oxygenase (i.e., Pden_1323), and this misleading adjective should be omitted from the text.

Likewise, that peptide removal from MP-11 was catalyzed by Ssr1698 was also not definitely shown. It is well known that bonds (i.e., thioether bonds to heme) can be cleaved in the mass spectrometer. Thus, that the free peptide was observed by mass spectrometry does not mean that it was generated enzymatically. Proper controls (e.g., on the MP-11 substrate standard or non-enzymatic oxidation product) could be used to confirm or rule out this claim. Additionally, the “conserved” HXXXDH motif bears minimal resemblance to the HXXEH motif of LuxS both in terms of sequence and location within the overall tertiary and quaternary structure of the enzyme, and thus does not provide compelling evidence that Ssr1698 catalyzes the cleavage of the thioether bonds of MP-11 in addition to heme oxygenase chemistry.

June 28, 2023

Professor Haichun Gao
Editorial Board Member
Communications Biology

Dear Professor Gao,

Thank you for handling our manuscript “Identification of a *c*-type heme oxygenase and its function during acclimation of cyanobacteria to nitrogen fluctuations” (COMMSBIO-22-3505A).

We appreciate reviewers’ constructive comments and have added a lot of new experiments (Fig. 3; Extended Data Fig. 3; Supplementary Figs. 7, 9 and 10) in the revised manuscript to strengthen our conclusion that Ssr1698 functions as a *c*-type heme oxygenase in cyanobacteria. Here we submit our revision of the paper with a point-to-point response to the reviewers’ comments.

Thank you very much for your help. We are looking forward to hearing from you about the revised manuscript.

Best regards,

Weimin Ma, PhD, Professor
College of Life Sciences
Shanghai Normal University
Shanghai 200234, China
Tel: 86-21-6432-1617 (Office)
E-mail: wma@shnu.edu.cn

Point-to-point response to reviewers' comments

Reviewer #1

As the authors have revised appropriately the manuscript according to the reviewers' comments, I recommend to accept this paper for the publication in Communications Biology.

Thanks!

Reviewer #2

The authors have addressed the majority of my comments, and I now support publication of this article in Communications Biology.

Thanks!

However, I have low confidence in the Alpha-Fold structure. The binding site of MP-11 in the revised manuscript is completely different from the binding site in the original manuscript. Ultimately, Alpha-fold is based upon homology modeling, and this is a novel protein with an unusual substrate MP-11. I don't think there is a reason to have high confidence in the Alpha-Fold structure. I suggest the authors considering removing this from the manuscript.

We agree with the reviewer's comments. We have deleted the Supplementary Fig. 8 (as shown in revised version 1) and its description obtained from the Alpha-Fold structure, but we still retain the structure as a model to indicate the binding stoichiometry of MP-11 (*c*-type heme) to Ssr1698.

Reviewer #3

Ran et al. made a number of revisions to their manuscript to address concerns regarding the characterization of a gene, *ssr1698*, that is upregulated during recovery from nitrogen-deficiency induced chlorosis via nitrate exposure. Concerns regarding artifactual effects due to the presence of a polyhistidine-tag on Ssr1698 were satisfactorily addressed. However, there are still many concerns that remain regarding the dual role of Ssr1698 as a novel *c*-heme oxygenase and phycobiliprotein maturase, and thus of its role in light-harvesting and nitrogen storage (i.e., the entire premise of the manuscript). These concerns are summarized below and should be strongly considered and/or addressed (either through experimentation or tempering of conclusions) if the manuscript is to be acceptable for publication.

To be honest, we are very grateful for the reviewer's constructive comments and suggestions. Following this reviewer's suggestions, we have added a lot of new experiments (Fig. 3; Extended Data Fig. 3; Supplementary Figs. 7, 9 and 10) in the revised manuscript to strengthen our conclusion that Ssr1698 functions as a *c*-type heme oxygenase in cyanobacteria.

It seems clear that the cell yield of *Synechocystis* sp. strain PCC 6803 increases upon nitrogen repletion and that deletion of *ssr1698* has a modest effect on the viability of this cyanobacterium in a light-dependent manner (e.g., Fig. 5d and Extended Data Fig. 5). However, there is contradictory data on whether repeated cycles of nitrogen depletion/repletion increases or decreases growth in response to the presence/absence of *ssr1698* (Fig. 5c vs. Extended Data Fig. 6).

Thanks for this good point. This comment reminds us to add a light intensity condition (growth light) in legends of Fig. 5 and updated Extended Data Fig. 5. Under growth light conditions, we observed that the cell growth is obviously decreased by *ssr1698* deletion during recovery from nitrogen chlorosis (Fig. 5a).

A significant decrease of cell growth caused by *ssr1698* deletion in repeated cycles of nitrogen depletion/repletion (Fig. 5c) but NOT in repeated cycles of only nitrogen repletion (updated Extended Data Fig. 5) is due to the fact that the *ssr1698* gene was preferentially expressed during recovery from nitrogen chlorosis but NOT under nitrogen repletion conditions (page 21, line 426 to page 22, line 431; page 23, lines 457 to 460). Therefore, the results of Fig. 5c and updated Extended Data Fig. 5 are not contradictory and they jointly support the conclusion that Ssr1698 is specific to operate under nitrogen oscillation conditions.

Likewise, it is unclear whether the enhanced photosystem and phycobiliprotein production upon nitrogen repletion is merely a byproduct of increased cell yield or even has anything to do with the presence of *ssr1698*. Specifically, it is unclear if the data in Fig. 1 have been properly normalized to the cell dry weight at each time point (the protein gels from low and high nitrogen conditions in Extended Data Fig. 1 do not appear to be). (Also, labels are missing for the traces in Fig. 1e.) Clearly, phycobiliproteins are still being produced in the Δ *ssr1698* strain at comparable levels to wild-type (Fig. 2d and 2e), so this gene is not essential for, or the likely primary source of, phycobilin production.

Thanks for this good point. Total protein in Fig. 1 has been normalized to 3×10^7 cells at each time point and NdhI was used as a sample loading control because the expression levels of NdhI are insensitive to conditions of low nitrogen and high nitrogen (Spät et al., 2018). Following this reviewer's suggestion, we have added the related information in legend of Fig. 1.

In Fig. 1e and Supplementary Fig. 2e, we utilized vertical axis lines of various colors

as labels. In order to avoid overlooking this information, we have added the related instructions in legends of Fig. 1e and Supplementary Fig. 2e.

Indeed, the deletion of *ssr1698* still leaves approximately half of the phycobiliproteins (Fig. 2d). However, the residual phycobiliproteins in the *ssr1698*-deletion mutant is unrelated to the assembly of intact phycobilisome with high molecular weight (Fig. 2f). Clearly, the expression of *ssr1698* gene is required for the synthesis and assembly of phycobiliproteins during recovery from nitrogen chlorosis.

Relatedly, it is not clear that phycobiliproteins are selectively used as a nitrogen reservoir. All proteins contain nitrogen and there is nothing in the data to suggest that phycobiliproteins are special in this regard over any other protein, save for their high abundance. Thus, it is odd at best and misleading at worst to highlight this feature, much less the role of *ssr1698* in modulating these effects as a mechanism of the nitrogen starvation response of *Synechocystis* sp.

Thanks for this good point. A large body of literature has reported that phycobiliproteins can be selectively used as a nitrogen reservoir in cyanobacteria (Wyman et al., 1985; Li et al., 2001; Forchhammer and Schwarz, 2019) because of their several unique properties. Compared to the photosystem proteins, phycobiliproteins are rich in nitrogen-containing amino acids, such as arginine, and have an exceptionally high nitrogen content at the same protein level (Krauspe et al., 2021, 2022; Wang et al., 2021), are much more abundant protein in cyanobacteria (Grossman et al., 1993), and exhibit remarkably rapid degradation and synthesis under conditions of nitrogen depletion and nitrogen repletion, respectively (Krasikov et al., 2012; Klotz et al. 2016; Spät et al., 2018).

In addition, the phycobiliproteins synthesized by *Ssr1698* under nitrogen repletion can account for up to 30% of the total soluble cell protein, as calculated from the results presented in Fig. 2d and reported by Gantt, 1981, and can function in ensuing nitrogen-starved cells as a nitrogen source. We propose that *Ssr1698* can modulate the nitrogen reservoir by synthesizing phycobiliproteins under nitrogen repletion, which is important for cell growth under nitrogen depletion.

Similarly, the data presented in the revised manuscript for *Ssr1698* as a novel c-heme oxygenase involved in phycobiliprotein maturation remains inconclusive. The UV-vis spectral changes taken as evidence for the 1:2 binding stoichiometry of both b- and c-heme to *Ssr1698* is not convincing. The effects of [*Ssr1698*] on the absorption spectrum of heme are subtle and controls should be used to rule out solvation effects or non-specific binding (e.g., by titrating in glycerol/PEG or BSA). The fact that the heme spectrum is heme-dependent, as stated in the text (line 358), does not provide evidence that *Ssr1698* binds heme.

Thanks for this good point. Following this reviewer's suggestion, we have utilized BSA as a control to rule out non-specific-binding. The results of the titration experiment indicated that *c*-type heme specifically binds Ssr1698 (see new Fig. 3b) and the binding stoichiometry of *c*-type heme to Ssr1698 is 1:2 (see new Fig. 3c). However, the binding of *b*-type heme to Ssr1698 is non-specific (see new Supplementary Fig. 7).

It is curious/concerning that the predicted binding site of *b*- and *c*-heme are not the same (Fig. 3e and Extended Data Fig. 3d), and that neither site is consistent with the site to which heme binds in HugZ (the heme-binding site of the DUF2470 domain of HugZ is on the opposite side of that predicted for the Ssr1698 monomer). That the proposed binding site of *c*-heme has polar residues (lines 462-467 and Supplementary Fig. 8) provides no evidence for whether or not Ssr1698 cleaves *c*-heme with α -meso regiochemistry (e.g., the polarity of this region could simply be a reflection of its solvent accessibility). Such regiochemistry would be necessary if Ssr1698 indeed was involved in the biosynthesis of phycobilin, but the authors have provided no such confirmation of this fact. Additional experiments (e.g., mass spec fragmentation patterns or comparison with biliverdin isomeric standards) would be needed to confirm the regiospecificity of Ssr1698, assuming it is a heme oxygenase at all.

Thanks for this good point. We agree with the reviewer's comments. We have deleted the Supplementary Fig. 8 (as shown in revised version 1) and its description obtained from the Alpha-Fold structure.

Following this reviewer's suggestion, we have analyzed the mass spectrometry fragmentation patterns. The identification of Ssr1698 as a *c*-type heme oxygenase is supported by the results showing that the mass spectrometry fragmentation pattern of the 583.3 *m/z* peak of the reaction product catalyzed by Ssr1698 is consistent with that of biliverdin IX α , a standard reference sample (new Supplementary Fig. 10).

This leads to the last and most critical point that the observed *c*-heme oxygenase activity of Ssr1698 has not been conclusively shown to be enzymatic. The observed spectral changes and time-course of "Ssr1698-catalyzed" MP-11 degradation (Fig. 4a) are reminiscent of those of HupZ (Traore et al, 2021), wherein it was concluded that this enzyme was likely NOT a heme oxygenase. This could be confirmed with proper no enzyme controls (for both *b*- and *c*-heme) and showing that the rate of the observed heme degradation activity scales linearly with [Ssr1698]. Regardless, the "Ssr1698-catalyzed" reaction is certainly not rapid (as claimed on line 380 and in the legend of Fig. 4), particularly when compared to a bona fide *c*-heme oxygenase (i.e., Pden_1323), and this misleading adjective should be omitted from the text.

Thanks for this good point. Following this reviewer's suggestion, we measured the rate of the observed *c*-type heme degradation activity with no enzyme control, and the results revealed a linear relationship with the concentration of Ssr1698 (new Extended

Data Fig. 3).

Following this reviewer's suggestion, we have removed the "rapidly" in the revised manuscript. Relevant revisions were made in page 20, line 389 and the legend of Fig. 4.

(Page 20, line 389), "The experimental data clearly showed that Ssr1698 lacked any observable reactivity with *b*-type heme (Supplementary Fig. 8), but had an ability of degrading the *c*-type heme (Fig. 4a)."

Likewise, that peptide removal from MP-11 was catalyzed by Ssr1698 was also not definitely shown. It is well known that bonds (i.e., thioether bonds to heme) can be cleaved in the mass spectrometer. Thus, that the free peptide was observed by mass spectrometry does not mean that it was generated enzymatically. Proper controls (e.g., on the MP-11 substrate standard or non-enzymatic oxidation product) could be used to confirm or rule out this claim. Additionally, the "conserved" HXXXDH motif bears minimal resemblance to the HXXEH motif of LuxS both in terms of sequence and location within the overall tertiary and quaternary structure of the enzyme, and thus does not provide compelling evidence that Ssr1698 catalyzes the cleavage of the thioether bonds of MP-11 in addition to heme oxygenase chemistry.

Thanks for this good point. Following this reviewer's suggestion, we have performed the mass spectrometry analysis of MP-11 (new Supplementary Fig. 9), a substrate for Ssr1698. The results indicated that the thioether bonds are cleaved by the Ssr1698 enzyme and not by the mass spectrometer.

We agree with the reviewer's comments. We have deleted the Supplementary Fig. S7 (as shown in revised version 1) and its description in text.

References

- Forchhammer K, Schwarz R. Nitrogen chlorosis in unicellular cyanobacteria—a developmental program for surviving nitrogen deprivation. *Environ Microbiol.* 2019; 21:1173–1184.
- Gantt E. Phycobilisomes. *Annu Rev Plant Physiol.* 1981; 32:327–347.
- Grossman AR, Schaefer MR, Chiang GG, Collier J. The phycobilisome, a light-harvesting complex responsive to environmental conditions. *Microbiol Rev.* 1993; 57:725–749.
- Klotz A, Georg J, Bučinská L, Watanabe S, Reimann V, Januszewski W, Sobotka R, Jendrossek D, Hess WR, Forchhammer K. Awakening of a dormant cyanobacterium from nitrogen chlorosis reveals a genetically determined program. *Curr Biol.* 2016; 26:2862–2872.
- Krasikov V, Aguirre von Wobeser E, Dekker HL, Huisman J, Matthijs HC. Time-series resolution of gradual nitrogen starvation and its impact on

- photosynthesis in the cyanobacterium *Synechocystis* PCC 6803. *Physiol Plant*. 2012; 145:426–439.
- Krauspe V, Fahrner M, Spät P, Steglich C, Frankenberg-Dinkel N, Maček B, Schilling O, Hess WR. Discovery of a small protein factor involved in the coordinated degradation of phycobilisomes in cyanobacteria. *Proc Natl Acad Sci USA*. 2021; 118:e2012277118.
- Krauspe V, Timm S, Hagemann M, Hess WR. Phycobilisome breakdown effector NblD is required to maintain cellular amino acid composition during nitrogen starvation. *J Bacteriol*. 2022; 204:e00158-21.
- Li H, Sherman DM, Bao S, Sherman LA. Pattern of cyanophycin accumulation in nitrogen-fixing and non-nitrogen-fixing cyanobacteria. *Arch Microbiol*. 2001; 176: 9–18.
- Spät P, Klotz A, Rexroth S, Maček B, Forchhammer K. Chlorosis as a developmental program in cyanobacteria: the proteomic fundament for survival and awakening. *Mol Cell Proteomics*. 2018; 17:1650–1669.
- Wang J, Wagner ND, Fulton JM, Scott JT. Diazotrophs modulate phycobiliproteins and nitrogen stoichiometry differently than other cyanobacteria in response to light and nitrogen availability. *Limnol Oceanogr*. 2021; 66:2333–2345.
- Wyman MR, Gregory RP, Carr NG. Novel role for phycoerythrin in a marine cyanobacterium, *Synechococcus* strain DC2. *Science*. 1985; 230:818–820.